# A gut microbiome-kidney-heart axis predictive of future cardiovascular diseases

Cardiovascular diseases (CVD) remain a major global health challenge. Early markers of disease initiation and progression are urgently needed. We, and others, have previously shown changes in the gut microbiome in association with metabolic and CVD. Here, we demonstrate that gut microbiome-related changes can be detected in association with subclinical variations in heart and kidney function. Markers related to gut microbial metabolism of aromatic amino acids, phenylalanine and tyrosine, associate with circulating pro-atrial natriuretic peptide and estimated glomerular filtration rate in a metabolically healthy European population. Observational and genetic evidence further identify microbiome-related metabolites as mediators of this gut microbiome-kidney axis, with their baseline levels associating with incident CVD in an external Canadian population. Altogether, our work suggests that the gut microbiome interacts with the cardiorenal axis and participates in an inter-organ crosstalk affecting host physiology and risk of CVD.

Dysmetabolism is the abnormal regulation of metabolic processes in the body, leading to imbalances in the way nutrients are processed, and energy is produced or utilized, and is often seen as an intermediary phase of the metabolic health-to-disease continuum. A dysmetabolic state is often characterized by the presence of one or more conditions like overweight, dyslipidaemia, hypertension or prediabetes, which are considered risk factors for cardiometabolic diseases (CMD) (including obesity and type 2 diabetes: T2D) and cardiovascular diseases (CVD). CVD, the leading cause of death and disability worldwide[1], are continually on the rise and are projected to hold the trend[2], unless underlying dysmetabolism can be targeted leading to a scenario forecasting a 13.3% reduction in global disease burden by 2050[2].

The gut microbiome is a key regulator of metabolism, with multiple reports linking it to metabolic health[3–5] and CVD[6,7]. These studies rely upon large population cohorts[4] and careful consideration of common confounding factors revealing microbiome-host phenotype[5] or disease signatures[6,7] with added confidence. While the description of a 'healthy' microbiome remains under scrutiny[8], recent work suggests that gut microbiome undergoes significant changes during dysmetabolism[6,7] with several of these changes persisting during CVD[6]. These observations suggest that the gut microbiome plays a key role both in the early stages of disease development and maintenance of the disease trajectory, thereby underscoring its untapped potential for prevention, early diagnoses and/or treatment of dysmetabolism, CMD and CVD.

Dysmetabolism, however, can represent a prolonged and metabolically heterogenous phase of CVD, therefore physiological variations underpinning disease initiation or progression can come from multiple mechanisms. Importantly, kidney function as an endogenous factor has been identified as a key contributor to the gut microbiome variation in healthy populations[5,9,10], whereas impairment of kidney function in advanced stages of CKD has been associated with gut dysbiosis[11]. In turn, a causative relationship between aberrant gut microbiota and kidney disease progression has also been suggested in humans[12], highlighting the underlying gut microbiome-kidney axis. Of note, kidney function and CVD are also tightly linked, such that even mild to moderate reduction in kidney function within the normal range is independently associated with an up-to 2-fold elevated risk of mortality due to CVD[13], and CKD is a prevalent co-morbidity in up to 60% of patients with heart failure[14]. This inter-relationship, often referred to as the cardiorenal or reno-cardiac syndrome, suggests that preventing kidney damage would also decrease associated cardiovascular morbidity and mortality[15] in advanced disease settings. However, whether kidney-heart interplay and its interactions with the gut microbiome play a role in disease initiation or early stages of cardiovascular pathophysiology remains unknown.

✉ e-mail: k.chechi@imperial.ac.uk; oluf@sund.ku.dk; stanislav.ehrlich@ucl.ac.uk; karine.clement@inserm.fr; m.dumas@imperial.ac.uk

Here, using plasma metabolomics and quantitative gut microbiome profiling[16] combined with extensive confounder-control in metabolically healthy controls (*n* = 275) of the MetaCardis study[6,17–19], we identify gut microbial markers related to estimated glomerular filtration rate (eGFR) and plasma pro-atrial natriuretic peptide (proANP) levels, revealing a gut microbiome-kidney-heart axis (Fig. 1). Briefly, this axis *i)* involves gut microbial metabolites and the host co-metabolites derived from phenylalanine and tyrosine metabolism, *ii)* is supported by genetic evidence indicating an interaction between the gut microbiome and kidney function of the host, the functional associations of which are depleted in the MetaCardis cases with CMD (*n* = 1602), and *iii)* associates with future CVD incidence in the participants of the Canadian Longitudinal Study on Aging (CLSA) study (n = 8,669). Our observations unravel an interplay between the gut microbiome and its host through plasma metabolites that are likely implicated in both the initiation and progression of CMD. Importantly, we identified these variations within the healthy group that are suggestive of trajectories towards metabolic dysregulation and CMD, which opens perspectives for precision prevention.

## Results

### Gut microbial and physiological characteristics of metabolically healthy individuals

To decipher phenome (i.e., clinical and anthropometric), plasma metabolome and gut microbiome variations in the early stages of metabolic health-to-disease continuum, we focused on the 275 metabolically healthy individuals (*i.e.*, individuals with BMI ≤ 25 kg/m², no metabolic syndrome, no T2D and no ischaemic heart disease (IHD)) in the MetaCardis[6,17–19] study, relative to the individuals with CMD (*i.e.*, presence of metabolic syndrome, obesity, T2D and/or IHD; *n* = 1602). These individuals were aged between 20 and 76 years (median age = 58 years), with 62% females and were recruited from Germany, France and Denmark (Supplementary Data 1). Expectedly, median values for the metabolic-health-related variables such as body fat distribution, lipid metabolism, glucose metabolism, blood pressure, liver and kidney function remained within the normal range in healthy individuals (Supplementary Data 1). However, 22.5% of these individuals had elevated blood pressure, 26.2% were prediabetic and 2–8% were medicated for conditions such as hypertension or received proton-pump inhibitors (Supplementary Data 1). The underlying variations in blood pressure, glucose and lipid metabolism in this group are likely indicative of their physiology transitioning from a healthy state towards dysmetabolism despite their metabolic disease-free status. Most gut microbiome features reflecting ecological or functional aspects – e.g., microbial load[16] i.e., microbial cells per gram of faecal matter), bacterial gene richness[20], and bacterial species richness[21] – remained comparable to previous reports in healthy individuals (Supplementary Fig. 1). Additionally, we observed low prevalence for the dysbiotic *Bacteroides 2* enterotype[22] among these individuals (only eight individuals, *i.e.* 2.9% prevalence), which has previously been linked to obesity[18], general dysmetabolism and IHD[6]. Consistent with previous reports[18], our healthy individuals with *Bacteroides 2* enterotype also exhibited strikingly lower microbial gene- and metagenomics species-richness relative to other enterotypes (Supplementary Fig. 1). Most of these (i.e., 84%) individuals also stated that they defecate regularly, which associated negatively with the prevalence of *Bacteroides 2* enterotype[16] (FDR < 0.1). Additionally, stool frequency of <1/2 per day associated with higher microbial load (FDR < 0.1) in these individuals, which is consistent with previous reports[16,23] (Supplementary Fig. 2a).

### Demographics as key microbiome covariates in metabolically healthy individuals

We examined the potential influence of host factors on the inter-individual variation in the gut microbiome composition and function of healthy participants (i.e., Bray-Curtis dissimilarity distance on genus,

metagenomic species (MGS), gut metabolic modules (GMM) and Kyoto Encyclopaedia of Genes and Genomes (KEGG) pathways, respectively). The analyses covered 144 available covariates including demographics (i.e., age, sex and country), diet (i.e., consumption of food groups and nutrients, overall caloric intake and key dietary scores including Alternative Healthy Eating Index (aHEI), Dietary Approaches to Stop Hypertension (DASH) and the Dietary Diversity Score (DDS)), other conditions (i.e., allergies, appendectomy, menopause, gout and obstructive sleep apnoea), medications, and phenomic- and stool-related variables.

In alignment with previous reports[9,10,24,25], host variables such as demographics, diet (i.e., yogurt, choline and folate consumption), digestive bloating, stool frequency and previous antibiotics intake associated with the compositional and functional variation in healthy individuals ($P_{univariate\ adonis}$ < 0.05; Supplementary Data 2), with demographics (Fig. 2a) and visceral fat mass remaining associated with metagenomics species variation at $FDR_{univariate\ adonis}$ < 0.1 (Supplementary Data 2). Demographics additionally remained consistent covariates of host microbiome (i.e., taxa and functions), metabolome and phenomic variables in individual analyses (Supplementary Figs. 2, 3, Supplementary Data 3), with age, sex (including menopause status) and country of participant recruitment influencing the gut microbiome at various taxonomic levels (Supplementary Figs. 4, 5, 6). Accordingly, we included corrections for demographics across our analyses and applied additional confounder-testing and control for individual microbiome-phenome associations using metadeconfoundR[6,17] as needed.

### Gut microbiome association with circulating proANP levels in metabolically healthy individuals

To evaluate the gut microbial correlates of a healthy physiological state transitioning towards dysmetabolism underscored by the hemodynamic variation noted in the healthy individuals, we employed demographics-adjusted univariate and multivariate analyses of microbiome-phenome associations. Notably, plasma concentration of creatinine and pro-ANP were among the top clinical variables that were explained ($R^2_{Yhat}$ = 0.26-0.48) and predicted ($Q^2_{Yhat}$ = 0.08-0.12) by multivariate covariate-adjusted partial least squares (CA-PLS)[26] models of gut microbiome composition and functions (i.e., MGS, GMM and KEGG pathways) in these individuals (Supplementary Data 4).

Similarly, besides replicating previous[20,27] observations such as lower abundances of *Roseburia*[27] and *Ruminococcaceae* genera associating with higher glycated haemoglobin levels (FDR < 0.1, Supplementary Fig. 7b) in univariate analyses, multiple gut microbial features also associated with kidney and heart function in these individuals (Fig. 2, and Supplementary Fig. 7). For instance, faecal microbial load (Fig. 2b) and the saccharolytic, proteolytic and lipolytic fermentation potential of the microbiome associated with higher plasma proANP concentration (FDR < 0.1, Supplementary Fig. 7a), whereas lower *Ruminococcus* and higher *Rothia* genera abundances associated with lower eGFR levels (FDR < 0.1, Supplementary Fig. 7b). In addition, lower species richness and abundance of *Bifidobacterium* associated with higher circulating sodium levels, whereas higher abundances of *Anaerotruncus* and *Pseudomonas* associated with higher circulating sodium levels in these individuals (FDR < 0.1, Supplementary Figs. 7a, b). Of note, alterations in the abundances of *Bifidobacterium, Anaerotruncus, Ruminococcus* and *Roseburia*, most of which are short-chain fatty acids (SCFA) producers, have been reported in association with CKD[28].

Intriguingly, plasma proANP concentration was also the only clinical variable that associated positively with gut microbiome functions in terms of multiple GMMs (78 out of 116) and KEGG pathways (168 out of 217) in healthy individuals (FDR < 0.1, Supplementary Figs. 7d and e, Supplementary Data 5). Notably, these proANP-associated gut microbial functions had multiple GMMs (e.g.,

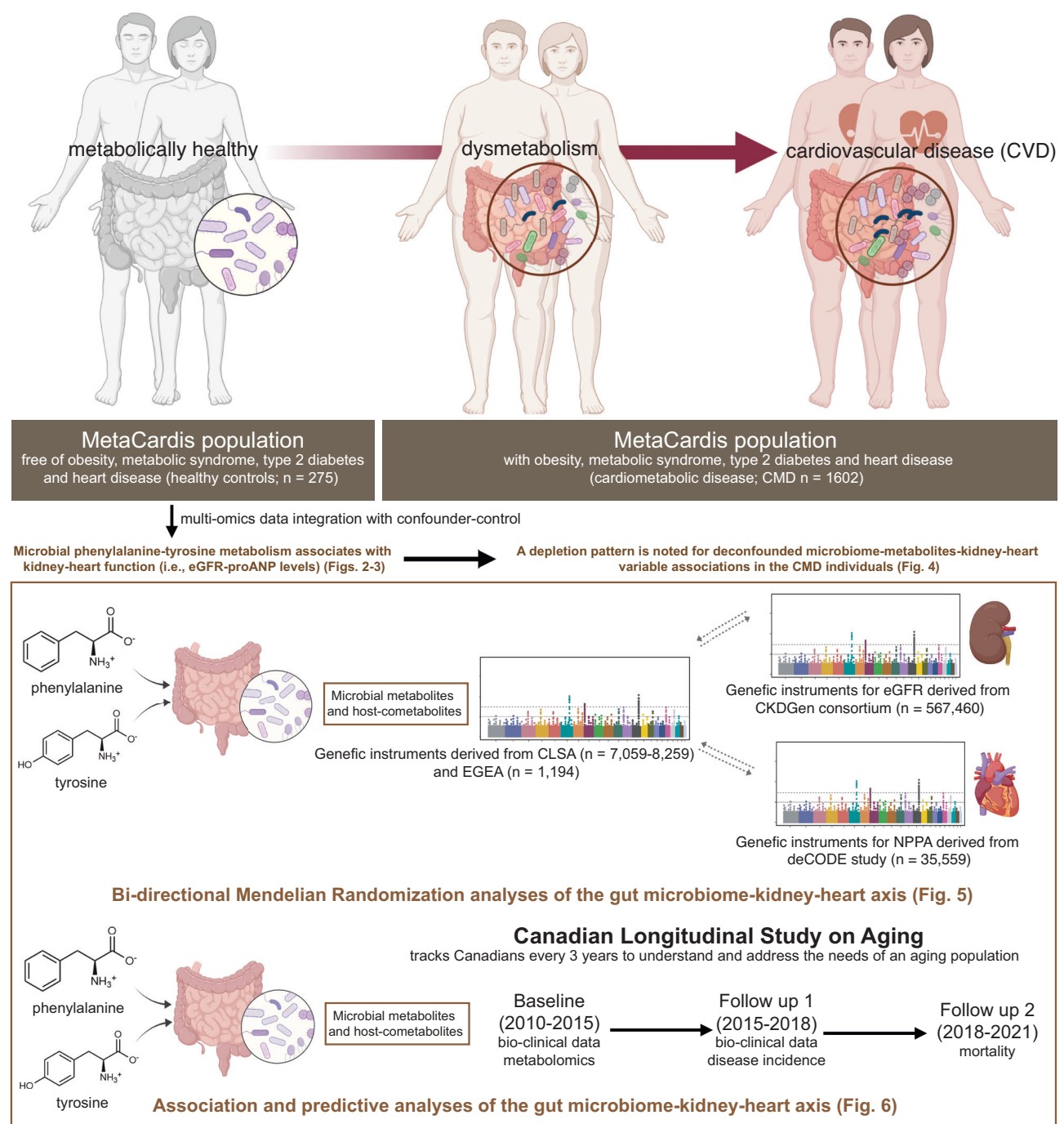

**Fig. 1 | Overview of the study.** Using integrative analyses of multi-omics data combined with extensive confounder-control, Mendelian Randomization (MR) and predictive analyses using multiple populations, we identify a gut microbiome-kidney-heart axis early during the dysmetabolic phase that is predictive of future CVD. We observe associations among microbial metabolism of phenylalanine and tyrosine with kidney and heart function, exemplified by eGFR and circulating proANP levels, respectively, in metabolically healthy controls of the MetaCardis study (i.e., free of metabolic syndrome, obesity, type 2 diabetes and IHD; $n = 275$; Figs. 2, 3), which exhibit a depletion pattern in CMD participants of the MetaCardis study (i.e., individuals with metabolic syndrome, obesity, diabetes and IHD; $n = 1602$; Fig. 4). Bi-directional MR analyses using genetic instruments derived from publicly available data from the CLSA, CKDGen and deCode studies for metabolites, eGFR and natriuretic peptide A, (NPPA) respectively, reveal a potential crosstalk between gut microbiome and host kidney function (Fig. 5). Finally, baseline levels of microbial metabolites and host co-metabolites derived from phenylalanine and tyrosine metabolism associate with kidney function and incident CVD in the CLSA study (Fig. 6) providing external validation of our findings. Created in BioRender. Chechi, K. (https://BioRender.com/p13eq37).

Tyrosine degradation II; Cinnamate conversion) and KEGG pathways (e.g., Phenylalanine metabolism; Tyrosine metabolism; Phenylpropanoid biosynthesis; Degradation of aromatic compounds) related to microbial aromatic amino acid metabolism (Fig. 2c, d).

Faecal microbial load was recently shown to be a key determinant of the gut microbiome variation and a confounder of microbiome-

disease associations[29]. As our microbiome profiles were quantitative in nature (i.e., data were adjusted for microbial load) and circulating proANP levels associated positively with microbial load (Fig. 2b), we next tested if the proANP-gut microbial functional associations were driven by faecal microbial load. All proANP-GMM and proANP-KEGG pathway associations were reducible to adjustment with microbial

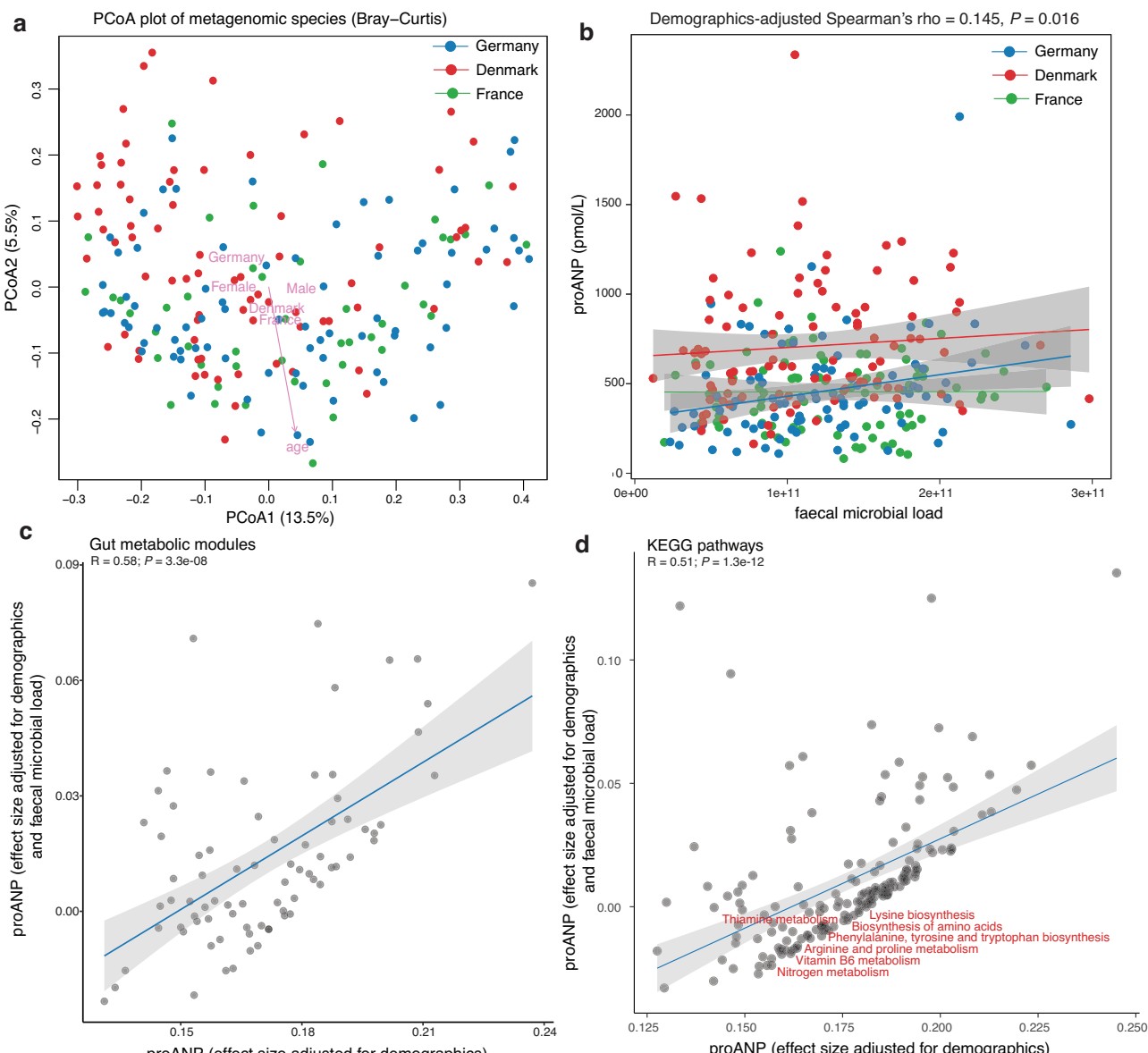

**Fig. 2 | A gut microbial functional signature for circulating proANP levels in healthy individuals. a** Principal coordinates analysis of inter-individual differences in microbiome profiles of metabolically healthy individuals of the MetaCardis study (MGS level Bray-Curtis dissimilarity index), coloured by country of recruitment (i.e., France, Germany and Denmark). Arrows and labels represent key host variables explaining variation in the gut microbiome composition ($n = 200$; MGS-$FDR_{univariateADONIS} < 0.1$; Supplementary Data 2). **b** Scatter plot showing positive association between circulating proANP levels and faecal microbial load in the MetaCardis healthy individuals ($n = 255$) tested using two-sided Spearman's rank correlation adjusted for demographics. Data show linear best-fit line ± 95% confidence intervals segregated for country of recruitment for visualization. Scatter plots showing associations among circulating proANP levels and absolute gut microbial functions at the level of (**c**) GMM and (**d**) KEGG pathways in the Meta-Cardis healthy individuals when adjusted for faecal microbial load (y-axes) in

addition to demographics (x-axes) ($n = 255$; multiple linear regression adjusted for demographics; multiple testing correction using BH-criteria, FDR < 0.1 considered significant). The effect sizes with and without faecal microbial load adjustment remained significantly correlated (Pearson's R, two-sided $p$ values and linear best-fit line ± 95% confidence intervals for visualization). While functional signatures of circulating proANP were reducible to faecal microbial load adjustment at FDR < 0.1 (Supplementary Data 6), seven KEGG pathways (shown in red) partially mediated the relationship between microbial load and circulating proANP levels in the healthy individuals (i.e., $FDR_{ACME} < 0.1$ Supplementary Data 8). MGS metagenomic species, GMM gut metabolic modules, proANP pro-atrial natriuretic peptide, BH Benjamini-Hochberg, FDR false-discovery rate, ACME average causal mediation effects, PCoA Principal coordinates analysis. Source data are provided as a Source Data file.

load (Fig. 2c and d, Supplementary Data 6), underscoring the value of quantitative microbiome profiling in microbiome studies. Seven of these 168 KEGG pathways, however, exhibited significant average causal mediation effects (ACME) ($FDR_{ACME} < 0.1$, Supplementary Data 7 and 8) when tested if any of above GMMs/KEGG pathway could mediate the relationship between microbial load and circulating proANP levels. These seven pathways predominantly included terms related to microbial amino acid metabolism (e.g., Phenylalanine,

tyrosine and tryptophan biosynthesis; Biosynthesis of amino acids; Supplementary Data 8).

As expected, plasma proANP levels also correlated positively with plasma creatinine levels and systolic blood pressure, and negatively with both visceral fat rating and total fat mass in these individuals (Supplementary Fig. 8a). Additionally, plasma proANP levels were significantly higher in individuals with hypertension relative to those with normal systolic blood pressure, and in individuals with

prediabetes relative to those with normal blood glucose levels within the metabolically healthy group (Supplementary Figs. 8b, c). However, none of the seven plasma proANP-KEGG pathways were confounded by hypertension or prediabetes status of the participants (Supplementary Fig. 8d).

Altogether, both univariate and multivariate analyses highlighted associations among gut microbiome and its functional features, especially those related to microbial amino acid metabolism, with circulating proANP levels in the healthy individuals.

### Circulating compounds of microbial aromatic amino acid metabolism associate with proANP in metabolically healthy individuals

To identify the plasma metabolites involved in the gut microbiome-proANP axis, we again used both demographics-adjusted univariate and multivariate approaches. Plasma metabolome-wide associations (i.e., 1484 metabolites) with circulating proANP levels revealed four metabolites with positive associations to plasma proANP: *N2,N2*–dimethylguanosine, malate, fumarate and vanillactate (FDR < 0.1; Supplementary Data 9) in the healthy individuals. *N2,N2*-dimethylguanosine is a purine nucleoside, while malate and fumarate are intermediates of the TCA cycle, and vanillactate is a putative microbial metabolite derived from the aromatic amino acid tyrosine. Circulating levels of these metabolites also increased with worsening kidney function (i.e., positively associated with creatinine and inversely with eGFR) (FDR < 0.1; Fig. 3a), suggesting that the metabolomics signature of proANP is intricately linked to the kidney-heart axis in these individuals.

We then employed multivariate CA-PLS models using gut microbiome features (i.e., MGS, GMM and KEGG pathways) as predictors to explain and predict individual metabolites while controlling for demographics. Again, several plasma metabolites derived from phenylalanine and tyrosine metabolism were among the top microbiome-predicted candidates (Fig. 3b, c, and Supplementary Data 10). As noted above, microbial KEGG pathway 'Phenylalanine, tyrosine and tryptophan biosynthesis' was also a key mediator linking microbial load with circulating proANP levels (Fig. 2d, and Supplementary Data 8), altogether suggesting a connection between proxies of kidney and heart function and the microbial metabolism of phenylalanine and tyrosine in these healthy individuals.

### Microbial phenylalanine and tyrosine metabolism associates with cardiorenal variables in metabolically healthy individuals

To further distil the role of microbial phenylalanine and tyrosine metabolism in kidney-heart physiology, we focused on the top plasma metabolites (i.e., well explained and predicted by the microbial taxonomic and functional features) of this pathway (Fig. 3b), which included 11 microbial metabolites and related host co-metabolites highlighted in Fig. 3c. A number of these metabolites, such as phenylacetylglutamine, phenylacetate, 3-(4-hydroxyphenyl)lactate and phenol sulphate, have previously been implicated in CVD[13], heart failure[30], metabolic dysfunction-associated steatotic liver disease[31,32] and diabetic kidney disease[33], and are considered uraemic toxins[34]. In contrast, low plasma 3-phenylpropionate levels were recently identified as a key marker of non-communicable multimorbidity[35].

These metabolites were highly correlated among each other (FDR < 0.1, Supplementary Fig. 9a) and exhibited widespread associations with the clinical and gut microbiome variables in the healthy individuals (Supplementary Figs. 9b-d, 10). Multiple metabolites including phenol sulphate, 3-(4-hydroxyphenyl)lactate, vanillactate and cinnamoylglycine associated with higher circulating levels of markers related to insulin and glucose metabolism (e.g., c-peptide, glycated haemoglobin, fasting insulin and free fatty acids) (FDR < 0.1, Supplementary Fig. 9b). Others, including phenol sulphate and vanillactate, associated with higher fasting triglycerides and liver gamma-glutamyl transferase as markers of lipid metabolism and inflammation,

respectively (FDR < 0.1, Supplementary Fig. 9b). Phenylacetylglutamine, 3-(4-hydroxyphenyl)lactate and vanillactate further associated with either low eGFR, high circulating creatinine levels (FDR < 0.1, Supplementary Fig. 9b) and/or higher circulating proANP levels. Altogether, these metabolites captured variations related to lipid, insulin and glucose metabolism as well as markers of kidney-heart function in the healthy individuals.

To address whether these metabolites could also capture underlying variations in the mitochondrial function, we tested their association with common circulating markers such as lactate, pyruvate, creatine and lactate-to-pyruvate ratio (i.e., lactate:pyruvate) using demographics adjusted linear models. Notably, phenylacetylcarnitine associated with higher circulating levels of creatin, whereas phenol sulphate, 3-(4-hydroxyphenyl)lactate and vanillactate associated with higher circulating levels of pyruvate or lactate (FDR < 0.1, Supplementary Fig. 9c), however, none of the mitochondrial function-related markers associated with proxies of kidney-heart function in the healthy individuals (FDR < 0.1, Supplementary Fig. 9d).

Plasma concentrations of 4-cresol, phenylacetate and related metabolites also associated with higher gene and species richness and species-based alpha-diversity of the microbiome (FDR < 0.1, Supplementary Fig. 9e) in alignment with previous reports[36–38], whilst also exhibiting widespread associations with individual genera, MGS, GMM and KEGG pathways in the healthy individuals, which overlapped with proANP-associated signatures, albeit not always in the same direction (Supplementary Figs 9f, and 10a–c). Additionally, association patterns of phenylacetate, 4-cresol and related host co-metabolites including phenylacetylglutamine, with the microbial fermentation potential ratios pointed towards an upregulation of proteolytic potential at the cost of both saccharolytic and lipolytic potential in these individuals (FDR < 0.1, Supplementary Fig. 9e). These observations link microbial proteolytic anaerobic metabolism and microbial amino acid metabolism with kidney-heart physiology in these individuals. Consistently, a higher microbial proteolytic-to-lipolytic ratio and processed-meat intake associated with lower eGFR and higher circulating creatinine levels in these individuals (FDR < 0.1, Supplementary Fig. 9g).

Given the widespread confounding evidenced earlier (Supplementary Fig. 2), we next evaluated whether the microbiome-metabolites-kidney-heart variables associations were confounded by host variables in the healthy individuals using the metadeconfoundR approach we previously introduced[6,17]. Of the 16,883 associations, 1375 were found to be deconfounded for the host variables listed in Supplementary Data 12 (Fig. 4a). Notably, the association of plasma proANP with vanillactate as a putative microbial metabolite was deconfounded (Supplementary Data 13). Likewise, inverse associations of eGFR with vanillactate, 3-(4-hydroxyphenyl)-lactate, phenylacetylglutamine and the genus *Rothia* were also deconfounded in the healthy individuals (Supplementary Data 13).

### Depletion of healthy gut microbiome-metabolites-kidney-heart associations in individuals with CMD

To determine if we had identified stable early markers of CMD driven by variations in kidney-heart function, we next examined these deconfounded microbiome-metabolite-kidney and heart variable associations in the metabolically unhealthy individuals of the Meta-Cardis study. This CMD population (n = 1602) included participants with metabolic syndrome, overweight or obesity (n = 682), T2D (n = 552) and IHD encompassing 111 acute coronary syndrome, 159 chronic IHD and 98 heart failure cases due to chronic IHD. As expected, the CMD group exhibited dysmetabolism, with higher BMI and enhanced central obesity, dyslipidemia, dysglycemia and elevated liver enzymes relative to the healthy group (Supplementary Data 1). Interestingly, however, plasma proANP levels were not significantly different between the two groups even after adjustment for demographics (Supplementary Fig. 11a).

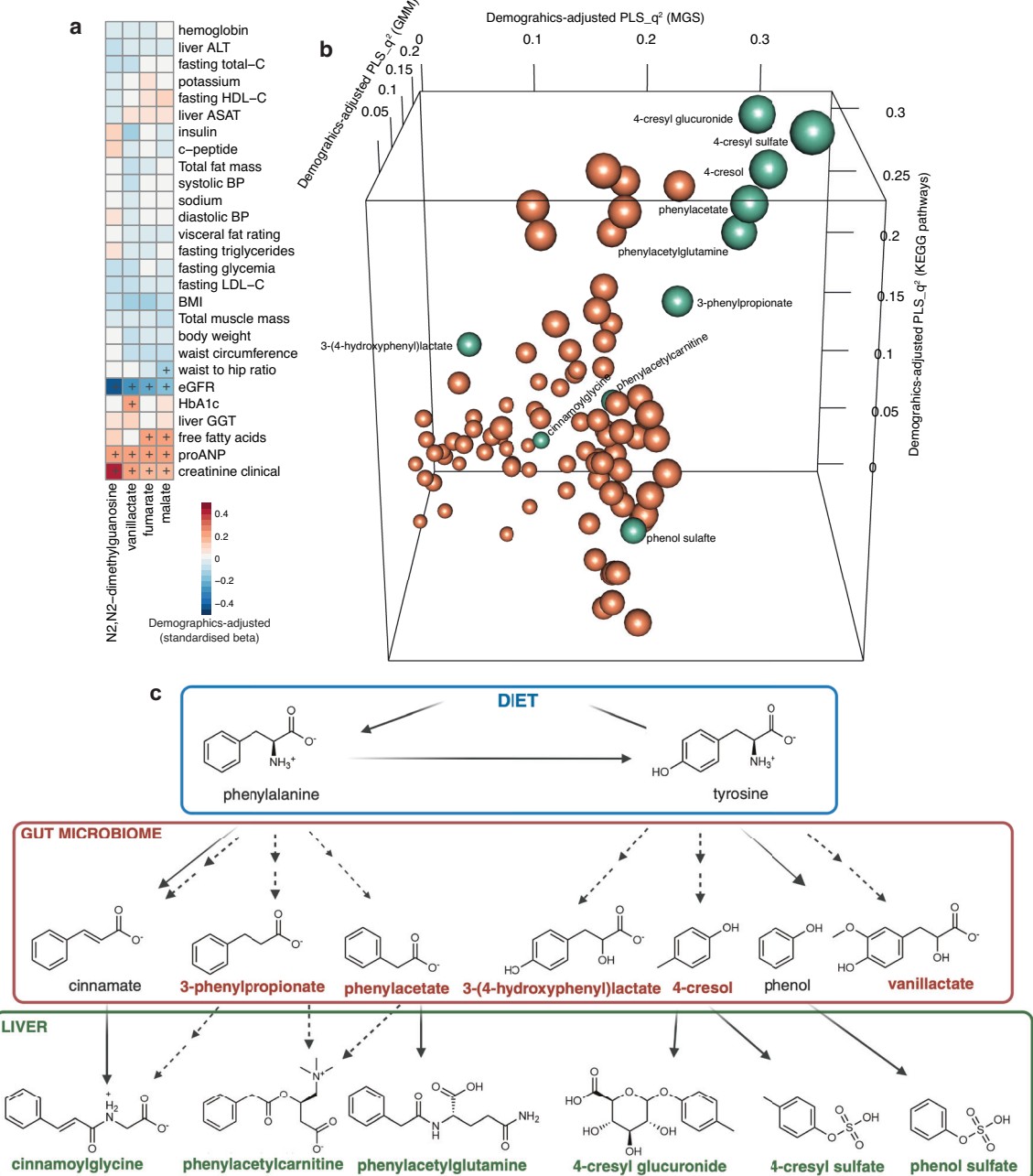

**Fig. 3 | Microbial metabolism of phenylalanine and tyrosine emerges as a key pathway in healthy individuals. a** Heatmap representing associations among metabolome and clinical phenotypes restricted to the metabolites exhibiting FDR < 0.1 for circulating proANP levels in the MetaCardis healthy individuals ($n$ = 254−274; exact sample sizes are given as Source data file; multiple linear regression adjusted for demographics; multiple testing correction using BH criteria, + represents FDR < 0.1). **b** 3-D plot representing top 50 robustly predicted metabolites (goodness-of-prediction, $q^2$) from multivariate CA-PLS models using gut microbiome taxonomic and functional features (i.e., MGS, GMM and KEGG pathways) as predictors and demographics (i.e., age, sex and country) as covariates in the MetaCardis healthy individuals (Supplementary Data 10) where point size represents maximum $q^2$ of all models and colour green represents the metabolites derived from phenylalanine and tyrosine metabolism versus the rest as brown. Key metabolites derived from microbial metabolism of phenylalanine and tyrosine labelled here were among the top metabolites that were explained and predicted by these models, which were built using five-fold cross-validations and 1000 Monte Carlo-based permutations. Metabolome was filtered for metabolites passing $q^2$ > 0 and FDR < 0.1 for each model (i.e., MGS, GMM and KEGG pathways) before identifying top 50 metabolites per category. **c** Overview of microbial (red) and host (green) metabolism of key metabolites derived from aromatic amino acids, phenylalanine and tyrosine investigated further in this study. Source data are provided as a Source Data file. CA-PLSq2, proportion of variance predicted (goodness of prediction) by covariate-adjusted partial least square regression models; MGS metagenomic species, GMM gut metabolic modules, proANP pro-atrial natriuretic peptide, ALT alanine aminotransferase, ASAT aspartate aminotransferase, GGT gamma-glutamyl transferase, HbA1c glycated haemoglobin, BP blood pressure, BMI body mass index, LDL-C low-density lipoprotein cholesterol, HDL-C high-density lipoprotein cholesterol, eGFR estimated glomerular filtration rate.

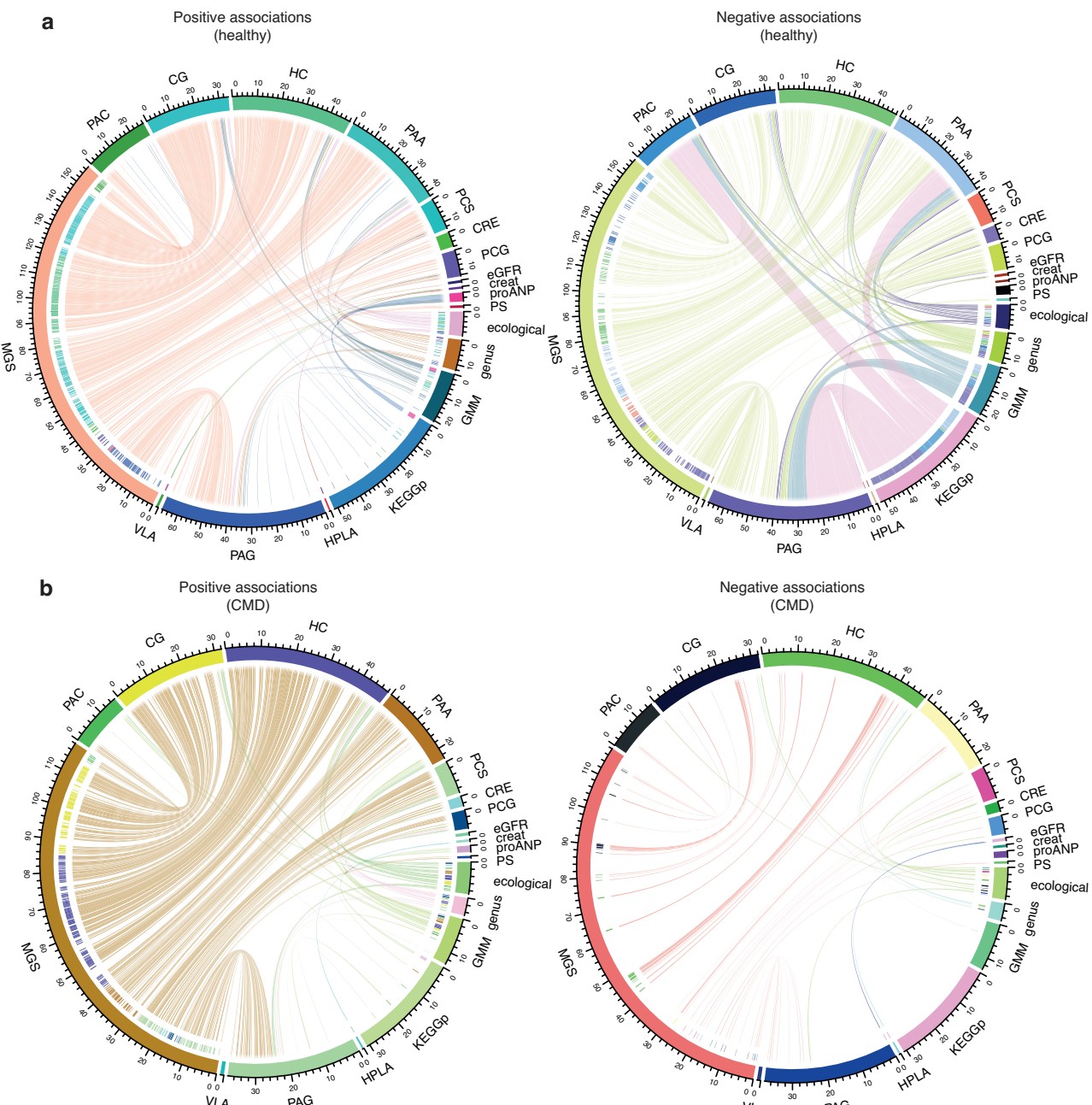

**Fig. 4 | Confounder-controlled gut microbial taxonomic and functional correlates of key phenylalanine and tyrosine metabolites in healthy individuals exhibit a depletion pattern in individuals with CMD.** Chord diagrams revealing significant inter-relationships among microbiome-metabolite-kidney-heart axis in (**a**) healthy individuals (*n* = 248–273) and (**b**) those with CMD (*n* = 1462–1575) (exact sample sizes are given in Supplementary Data 13) in the MetaCardis population. Inter-relationships are segregated for positive (left) and negative associations (right). Gut microbiome features included higher ecological, compositional (i.e., genus and MGS) and functional (i.e., GMM and KEGG pathways) aspects, which served as predictors in models explaining key metabolites and kidney-heart variables (i.e., proANP, eGFR, creatinine), whereas metabolites only served as predictors in models explaining kidney-heart variables. Inner rings of the chord diagrams correspond to the colour of the response variable (i.e., predicted metabolite or clinical variable). Connections of the chord diagrams represent significant inter-relationships coloured by the predictor variable where gut microbial features were grouped as ecological, genera, MGS, GMM and KEGG pathways. Specifically, demographics-adjusted Spearman's rank correlation was used for higher

ecological, genus and MGS features whereas demographics-adjusted multiple linear regression using rank-normalized data was used for GMM, KEGG pathways and metabolites. Multiple testing correction was done using BH-criteria and FDR < 0.1 was considered significant. Additionally, metadeconfoundR analysis was employed to test all possible confounders including diet, medication, other conditions, stool and lifestyle related variables covered in Supplementary Data 12. Associations not confounded by any of these variables are plotted in **a** and only associations identified in the healthy individuals that replicated in the CMD individuals are plotted in **b**. Replication was ascertained by 1. statistical significance, 2. directional confluence with healthy and 3. deconfounded status in the CMD individuals (Supplementary Data 13). CRE, 4-cresol; PCS, 4-cresyl sulphate; PCG, 4-cresyl glucuronide; PAA phenylacetate, PAG phenylacetylglutamine, PAC phenylacetylcarnitine, HC, 3-phenylpropionate, CG cinnamoylglycine, PS phenol sulphate, HPLA, 3-(4-hydroxyphenyl)-lactate, VLA vanillactate, proANP pro-atrial natriuretic peptide, eGFR estimated glomerular filtration ratio calculated according to MDRD, MGS metagenomic species, GMM gut metabolic modules, KEGG pathways, KEGG pathways, creat creatinine.

Circulating levels of key metabolites were also significantly different between healthy and CMD groups, with phenol sulphate, phenylacetylcarnitine and 3-(4-hydroxyphenyl)lactate being significantly higher, and cinnamoylglycine and 3-phenylpropionate being markedly lower in the CMD group relative to the healthy individuals (demographics-adjusted ANCOVA, $P < 2e-16$; Supplementary Fig. 11b). Interestingly, these metabolites also exhibited widespread associations with kidney and heart-related variables including left ventricular ejection fraction (LVEF) in the CMD, even when they were split for their disease category, with the associations becoming stronger with advancing CMD (FDR < 0.1, Supplementary Fig. 11c), highlighting the interconnectedness of kidney-heart function with disease progression.

Expectedly, we noted inverse associations between eGFR and circulating proANP levels[39] ($P_{healthy} = 0.11$, $P_{CMD} = 1.26e-15$), positive associations between eGFR and LVEF ($P_{healthy} = 0.48$, $P_{CMD} = 0.0014$) and inverse associations between circulating proANP levels and LVEF ($P_{healthy} = 0.93$, $P_{CMD} = 4.51e-15$), in both healthy and CMD groups; however, statistical significance was only achieved in the CMD individuals (demographics-adjusted linear models; Supplementary Fig. 11d).

We then examined the deconfounded associations between gut microbiome features, plasma metabolites and kidney-heart variables in the CMD group. Applying the stringent criteria of 1) statistical significance at FDR < 0.1, 2) deconfounded status for the covariates listed in Supplementary Data 12 including additional medications for CMDs and 3) effect size directional alignment with the healthy controls, we found that only 472 out of 1375 associations (Fig. 4a) replicated in the CMD group with a clear non-replication of inverse associations (Fig. 4b, and Supplementary Data 13).

In healthy individuals, we observed dense patterns of deconfounded associations among gut microbiome features with phenylacetylglutamine, followed by phenylacetate, 3-phenylpropionate, cinnamoylglycine and phenylacetylcarnitine, and then by 4-cresyl glucuronide, 4-cresyl sulphate and 4-cresol (Fig. 4a). Interestingly, many of these associations were inverse with a notable over-representation of gut microbial functions (i.e., GMMs and KEGG pathways) associating inversely with phenylacetylglutamine, phenylacetate and phenylacetylcarnitine. Notwithstanding the bi-directionality and scope of the coverage of overall gut microbiome functional potential by our features, these observations suggest that the gut microbiome in healthy individuals is functionally geared towards maintaining the circulating levels of these potentially deleterious metabolites within the homoeostatic range. Conversely, the apparent loss of most inverse associations between gut microbiome features and our metabolites of interest in the CMD group could likely be due to utilization of alternate metabolic pathways, competitive exclusion of metabolite-lowering bacteria, enhanced host-microbe interactions for metabolite retention or microbial adaptation to diet with CMD development. Altogether, we observed a lack of replication of the plausibly beneficial associations in the CMD group (Fig. 4b), which may further imply that CMD development is likely associated with loss of 'healthy' gut microbial features. Key examples of such depletion include abundance of butyrate-producing genera *Roseburia* and *Faecalibacterium* and related MGS *Faecalibacterium prausnitzii*, *Roseburia faecis* and *Roseburia inulinivorans* known for their metabolic-health-promoting roles[40,41], which associated inversely with either one or all of phenylacetate, phenylacetylglutamine, phenylacetylcarnitine, 4-cresyl sulphate, 4-cresyl glucuronide and 4-cresol levels in the healthy individuals and were non-significant in the CMD cases (Supplementary Data 13). Another SCFA-producing genus, *Dorea*, which associated inversely with vanillactate in our healthy individuals and is reduced in people with renal dysfunction[42], was also unreplicated in the CMD cases. In contrast, genus *Rothia* that exhibited inverse associations with eGFR in healthy individuals while associating positively with phenylacetate, phenylacetylglutamine, 4-cresyl sulphate and 4-cresyl glucuronide retained significance with

phenylacetate in CMD cases (Supplementary Data 13). These findings indicate that the gut microbiome's interactions with kidney and heart function through key metabolites can change during disease development. This aligns with previous observations of an overall depleted gut microbial and metabolomics pattern in myocardial infarct cases relative to healthy controls in an Israeli population[7]. In the same study, a gut microbial genome–exhibiting significant associations with higher circulating levels of phenylacetylglutamine, phenylacetate and 4-cresyl glucuronide–was also depleted in the cases with myocardial infarction[7].

## Microbial phenylalanine and tyrosine metabolites mediate the gut microbiome-kidney-heart axis

To decipher the plausible dependencies among our deconfounded gut microbiome features, plasma metabolites and kidney-heart variable associations, we next conducted mediation analyses of the microbiome on eGFR (Fig. 5a) and plasma proANP (Fig. 5b) in the healthy individuals. We found plasma cinnamoylglycine, phenylacetylglutamine, 4-cresyl sulphate and vanillactate to be the mediators of gut microbiome-eGFR associations ($FDR_{ACME} < 0.1$; Fig. 5a, and Supplementary Data 14), where indirect effects were noted at all levels of the gut microbiome (i.e., ecological, compositional and functional). For instance, genus *Dorea* associated with higher eGFR mediated by lower circulating vanillactate levels, whereas genus *Rothia* associated with lower eGFR via higher circulating phenylacetylglutamine levels (Supplementary Data 14). Similarly, *Faecalibacterium*, *Bacteroides*, and *Roseburia*, all SCFA-producing genera, and related species *Faecalibacterium prausnitzii*, *Roseburia faecis* and *Roseburia inulinivorans* associated with higher eGFR via lower circulating levels of phenylacetylglutamine and 4-cresyl sulphate. In contrast, multiple *Clostridium* spp., previously reported to be enriched in CKD[2], and genus *Victivallis* associated with lower eGFR via higher circulating levels of cinnamoylglycine (Supplementary Data 14). Conversely, the gut microbiome exhibited only direct effects on circulating proANP levels (i.e., average direct effects (ADE); $FDR_{ADE} < 0.1$, Fig. 5b, Supplementary Data 15).

Our analyses revealed that a higher propensity for proteolytic fermentation, relative to both lipolytic and saccharolytic fermentation (i.e., higher proteolytic-to-lipolytic ratio and lower saccharolytic-to-proteolytic ratio), as well as multiple gut microbial functional pathways (i.e., GMM and KEGG pathways) related to phenylalanine and tyrosine metabolism, were associated with lower eGFR through elevated circulating levels of phenylacetylglutamine, 4-cresyl sulphate, and cinnamoylglycine (Supplementary Data 14). Consistently, a higher capacity for phenylalanine and tyrosine metabolism in conjunction with higher circulating levels of uraemic toxins (i.e., 4-cresyl sulphate and phenylacetylglutamine) has previously been reported in CKD patients[12]. Of note, these microbial pathways remained associated with higher circulating proANP levels via direct effects ($FDR_{ADE} < 0.1$, Fig. 5b, and Supplementary Data 15) in the healthy individuals, indicating a role for microbial phenylalanine and tyrosine metabolism in regulating circulating proANP levels beyond our tested metabolites. Collectively, our mediation analyses identified clear dependencies among gut microbial metabolism of phenylalanine and tyrosine and kidney-heart variables in these healthy individuals.

Our plasma mediator metabolites are well-known uraemic toxins suggesting that they can also be regulated by kidney function. Thus, we next employed Mendelian Randomization (MR)[43] as an instrumental variable method to address 1) the inter-relationships among our plasma metabolites and the kidney-heart variables (i.e., eGFR and proANP) in both directions, and 2) to triangulate our observations using genetic evidence. We derived genetic instruments for these metabolites, eGFR and Somascan aptamer targeting circulating natriuretic peptide A (NPPA) from the CLSA[44,45] ($n = 7059-8029$, Supplementary Data 16), CKDGen consortium[46] ($n = 567,460$) and deCODE study[47] ($n = 35,559$), respectively. Additionally, we used the

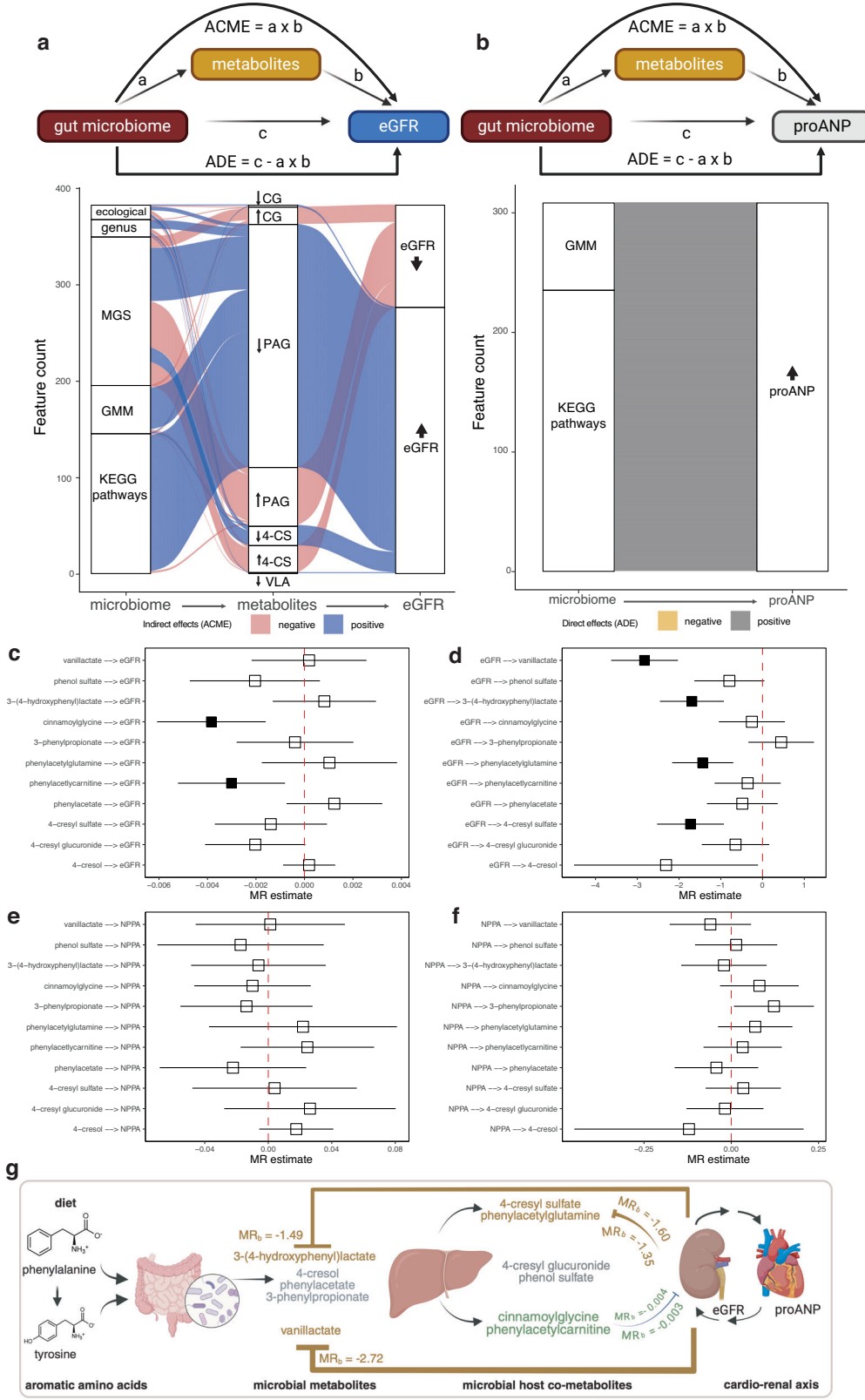

Epidemiological study on the Genetics and Environment of Asthma (EGEA)[48] ($n = 1194$) to derive the genetic instruments for circulating 4-cresol levels (Supplementary Data 16).

Genetically predicted circulating levels of cinnamoylglycine associated inversely with genetically predicted levels of eGFR in our initial univariate inverse-variance weighted (IVW) analysis

(FDR$_{IVW}$ < 0.1, Supplementary Data 17), which remained robust to additional methods testing various MR assumptions. Removing genetic instruments exhibiting significant heterogeneity (as measured by Cochrane's $Q < 0.05$ and $I^2$ statistics $\geq 25\%$) further validated an inverse association between genetically predicted levels of cinnamoylglycine and eGFR, while additionally identifying an inverse

**Fig. 5 | Mediation and Mendelian Randomization analyses reveal the gut microbiome-kidney-heart axis.** Alluvial plots showing (**a**) ACME or the indirect effects of the gut microbiome features on eGFR via key microbial metabolites as mediators and (**b**) ADE or the direct effects of the gut microbiome features on circulating proANP levels derived from demographics-adjusted mediation analyses using two-sided significance testing in the MetaCardis healthy individuals ($n = 230-271$, exact sample sizes in Supplementary Datas 14 and 15). While the indirect effects of each microbiome-metabolite association are coloured for the overall ACME on eGFR, microbiome-mediator associations were separated into positive or negative based on their effect sizes shown in Supplementary Data 13. Bidirectional relationships among key microbial metabolites, eGFR and proANP were next investigated using the two-sample MR testing. Forest plots show IVW estimates ± 95% confidence intervals for analyses involving (**c**) key metabolites as exposures and eGFR as outcome (Supplementary Data 17), **d** eGFR as exposure and metabolites as outcomes (Supplementary Data 18), **e** key metabolites as exposures and NPPA as outcome (Supplementary Data 19) **f** and NPPA as exposure and metabolites as outcomes (Supplementary Data 20). Number of genetic instruments

($n$) given in respective Supplementary Data. Filled squares represent significance at $FDR_{IVW} < 0.1$ and robust analyses that passed additional sensitivity tests. In cases, where significant heterogeneity (as measured by Cochrane's $Q < 0.05$ and $I^2$ statistics $\geq 25\%$) was observed, IVW estimates post outlier correction are shown. **g** Schematic summarizing the MR findings providing genetic evidence for two-way relationships among microbial metabolites and their host co-metabolites, eGFR and NPPA revealing the gut microbiome-kidney-heart axis. Arrows (pointed or blocked) represent stimulating versus inhibitory effects, respectively, whereas thickness of the connections represents strength of the MR estimates. Only associations exhibiting robust MR estimates that passed additional sensitivity analyses criteria are shown. Created in BioRender. Chechi, K. (https://BioRender.com/3amltcg). ACME mediation effects, ADE average direct effects, eGFR estimated glomerular filtration ratio, NPPA natriuretic peptide A, IVW inverse-variance weighted, MR Mendelian Randomization. CG cinnamoylglycine, PAG phenylacetylglutamine, 4-CS 4-cresyl sulphate, VLA vanillactate, MGS metagenomics species, GMM gut metabolic modules.

association between genetically predicted levels of phenylacetylcarnitine and eGFR ($P_{IVW-cinnamoylglycine} = 0.007$; $P_{IVW-phenylacetylcarnitine} = 0.008$; Fig. 5c; and Supplementary Data 17).

Next, we tested the influence of genetically predicted host kidney function on circulating levels of these metabolites. Consistent with our MetaCardis observations, genetically predicted eGFR levels associated inversely with genetically predicted circulating levels of vanillactate, 3-(4-hydroxyphenyl)lactate, phenylacetylglutamine and 4-cresyl sulphate ($FDR_{IVW} < 0.1$, Fig. 5d, and Supplementary Data 18), findings which were robust to additional sensitivity analyses testing violations of MR assumptions and instrument heterogeneity.

MR investigations of the metabolite-NPPA inter-relationships did not yield any significance (Fig. 5e, f, and Supplementary Data 19–20), whereas in alignment with the reported physiological effects of ANP on increasing GFR[49], genetically predicted levels of NPPA associated positively with eGFR ($P_{IVW} = 0.035$) (Supplementary Data 21). Post outlier correction, however, both eGFR and NPPA were positively associated in each direction (NPPA-eGFR: $P_{IVW} = 0.023$; eGFR-NPPA: $P_{IVW} = 0.008$) and only robust to additional sensitivity analyses in the eGFR-NPPA direction (Supplementary Data 21).

Altogether, the MR analyses validated our mediation analyses by adding genetic evidence to suggest that microbial metabolites can potentially shape host kidney function. In addition, these data suggest a putative role for host kidney function in regulating circulating levels of key microbial metabolites commonly described as uraemic toxins (Fig. 5g).

## Microbial phenylalanine and tyrosine metabolites associate with CVD incidence

Finally, to determine the clinical relevance of our key plasma metabolites, we evaluated their predictive potential in independent longitudinal studies. The CLSA[44] study is following 51,338 Canadians, aged 45–85 years at enrolment, recruited from 2010 to 2015 and being tracked longitudinally every 3 years, to understand and address the needs of an aging population. We used their baseline plasma metabolomics data ($n = 8669$) to test if our key metabolites 1) associate with eGFR at baseline and shifts in kidney function during first follow-up (i.e., 2015–2018, FU1), 2) associate with CVD incidence during FU1 and 3) add predictive value to the traditional clinical markers[6] of CVD. To expand the timeframe of these observations, we looked up the publicly available webserver by Pietzner et al. [35]., where plasma metabolite-disease associations are available for 11, 966 Europeans in the EPIC-Norfolk cohort[35] with a ~ 20-year follow-up period.

Age- and sex-adjusted linear mixed models revealed that all plasma metabolites except 3-phenylpropionate associated with

baseline eGFR, with vanillactate exhibiting largest effect sizes followed by phenylacetylglutamine, 3-(4-hydroxyphenyl)lactate, 4-cresyl sulphate and others in CLSA consistent with our observations in both MetaCardis and population studies used for MR analyses (Fig. 6a). These patterns also remained consistent in the EPIC-Norfolk data where both eGFR and prevalent kidney disease explained most of the variance in the circulating levels of 3-(4-hydroxyphenyl)lactate (4%, 7%), followed by phenylacetylglutamine (4%, 5%) and 4-cresyl sulphate (2.5%, 2.5%), respectively (Supplementary Fig. 12a), further validating our observations of microbiome-derived metabolites and eGFR interactions. Average eGFR in CLSA ($76.1 \pm 14.9$) shifted modestly over the 3-year FU1 period ($73.9 \pm 15.8$); nonetheless, key metabolites including vanillactate also associated with this change in CLSA (Supplementary Fig. 13).

Considering this impact, we next employed age-, sex- and eGFR-adjusted Cox-proportional hazards models, which revealed multiple associations among our key plasma metabolites and disease incidence in CLSA during FU1. Notable examples include 4-cresyl sulphate, 4-cresyl glucuronide, phenylacetylglutamine, 3-(4-hydroxyphenyl)-lactate and vanillactate, baseline levels of which were associated with a higher incidence of myocardial infarction during the next three years (FDR < 0.1, Fig. 6b). Among these, 4-cresyl glucuronide, phenylacetylglutamine, and vanillactate were also linked to increased overall mortality (FDR < 0.1, Fig. 6b). We also noted baseline levels of 3-phenylpropionate associating with lower mortality in alignment with previous observations[35] in addition to its inverse associations with CKD during FU1 (FDR < 0.1, Fig. 6b). Altogether, baseline levels of most of our key metabolites associated with future incidence of CVD, in particular myocardial infarction, which replicated for all metabolites, except 4-cresyl sulphate, in terms of future coronary heart disease incidence in the EPIC-Norfolk data (FDR < 0.1; Supplementary Fig. 12b).

Lastly, we built machine-learning models using multivariate Cox-regression where the CLSA-FU1 population was randomly split into 70/30, where the 70% set was used for training with nested 5-fold cross-validations, followed by testing in the held-out 30% test set. These models were bootstrapped 1000 times to improve stability and minimize overfitting. We compared three models considering 1) routinely screened clinical markers relevant to CVD (i.e., age, sex, body mass index, systolic blood pressure, glycated haemoglobin (factored as >5.7, 5.7–6.4 and <6.4%), smoking status, fasting plasma concentrations of LDL-cholesterol, HDL-cholesterol, triglycerides and eGFR), 2) six plasma metabolites with genetic evidence (i.e., phenylacetylcarnitine, cinnamoylglycine, phenylacetylglutamine, 4-cresyl sulphate, vanillactate and 3-(4-hydroxyphenyl)lactate), and 3) a combination of the two as predictors and disease incidence as outcome in CLSA. Notably, inclusion of these metabolites significantly improved the predictive

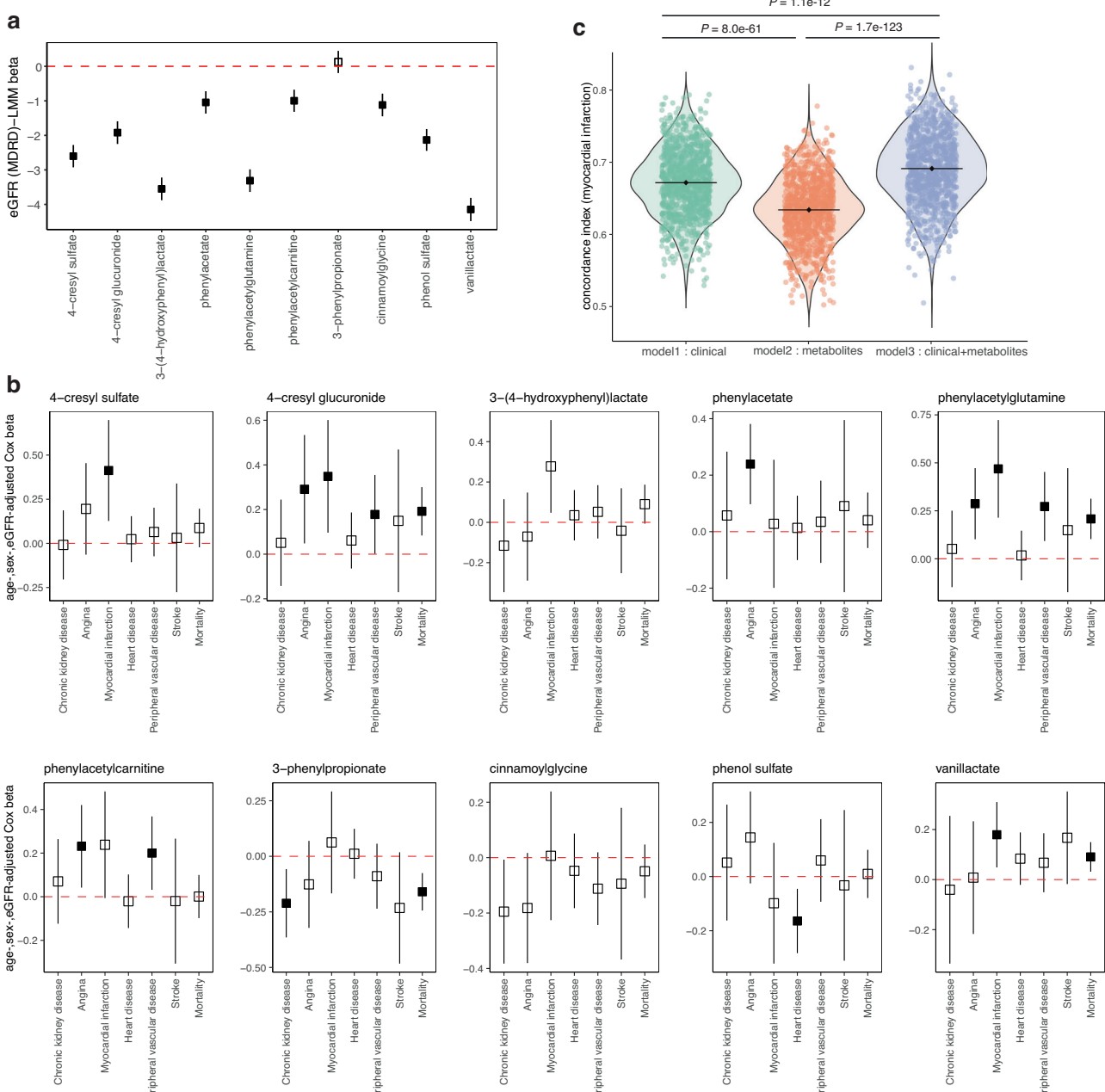

**Fig. 6 | Key metabolites derived from phenylalanine and tyrosine metabolism associate with incident CVD in a Canadian population. a** Forest plot showing associations among key metabolites measured at baseline and eGFR in the CLSA study population at two time points i.e., baseline and FU1 in 8669 individuals. Data are shown as β estimates ± 95% CIs derived from linear mixed models (LMM) adjusted for age, sex and time (days since baseline), using metabolites as exposures (log-transformed and standardized) and an interaction term across two time points. Filled squares represent significance at FDR < 0.1 for ten tested metabolites. **b** Forest plots representing associations among key metabolites measured at baseline and disease incidence during the FU1 period in the CLSA study and mortality data available till FU2 period. Data are shown as β estimates ± 95% CIs derived from age-, sex- and baseline eGFR- adjusted Cox proportional hazards models with interval-censoring (and right-censoring for mortality data) using baseline metabolite levels (log-transformed and standardized) as exposures and each disease incidence at FU1 as a binary variable (cases at baseline were excluded) considering time since baseline (days) in the CLSA cohort. Filled squares represent significance at FDR < 0.1 for seven tested conditions per metabolite. **c** Violin plots comparing predictive accuracy (measured as concordance index) of the models predicting myocardial infarction incidence during FU1 in the CLSA study using multivariable Cox proportional hazards regression models with interval-censoring employing baseline clinical and metabolomics data. Model 1 included clinical variables relevant to heart disease, model 2 included six key metabolites with MR evidence and model 3 included both clinical and metabolites as predictors. Predictions were made using models randomly splitting the CLSA population into training (70%) and test (30%) sets, followed by five-fold cross-validations per training set and testing using the held-out test set, with 1000 bootstraps. Data are shown as violin distributions with individual points and their respective median. *P* values were derived from Kruskal-Wallis test with Dunn-Bonferroni *post hoc* tests. eGFR, estimated glomerular filtration ratio calculated according to MDRD formula. Source data are provided as a Source Data file.

accuracy (i.e., concordance-index) relative to the model based on clinical features alone for predicting myocardial infarction (Fig. 6c) and mortality (Supplementary Fig. 14), implicating these metabolites in IHD development. Altogether, these observations underscore that the gut microbial metabolites derived from phenylalanine and tyrosine metabolism are clinically relevant and hold predictive value suggesting that any future interventions targeting these metabolites or the underlying gut microbial drivers might intercept CMD and CVD progression.

## Discussion

In multicellular organisms, the gut microbiome has co-evolved as a metabolic and signalling organ shaping host physiology via regulation of organ development and function. Indeed, the interactions between gut microbiome and various host organs[50] and related inter-organ axes[51,52] are increasingly being recognised for their role in health and disease; however, most of these remain to be supported by evidentiary proof. The natural history of chronic CVD also highlights a gradual failure of multi-organ cross talks, which likely underpins the transition from an asymptomatic to a diseased state. Given its role in regulating physiological and pathophysiological inter-organ communication, the gut microbiome is likely to play a significant role in this process.

Here, in the European MetaCardis cohort ($n = 1877$) we provide epidemiological evidence on the existence of a gut microbiome-kidney-heart axis that is detectable in the early stages of metabolic dysregulation present in healthy individuals ($n = 275$); findings that were extrapolated to individuals with overt metabolic disease and IHD (CMD, $n = 1602$). Our observations illustrate a key role for the gut microbial metabolism of aromatic amino acids, phenylalanine and tyrosine, and related metabolites in shaping human cardiovascular health. These microbial metabolites and their host co-metabolites are suggested to be key mediators of the gut microbiome-peripheral organ crosstalk, which associate with future CVD incidence as shown in the CLSA cohort.

We captured a gut microbial signature for the variations in kidney-heart function (i.e., eGFR-plasma proANP levels) in our metabolically healthy individuals who were controls for the participants with CMD (i.e., metabolic syndrome, obesity, T2D and IHD) in the MetaCardis study. However, exclusion of diseases does not result in a population devoid of risk factors[9], as 40% of these individuals had either hypertension or prediabetes (8% had both) both linked with an early impairment of kidney function as independent risk factors. Additionally, T2D and hypertension were identified as the two major causal factors driving the increasing burden of CKD and related CVD mortality globally[53]. Indeed, the relationship between impaired kidney function and heart disease is well-established with studies reporting increasing mortality due to CVD as eGFR declines[54,55]. We also noted an inverse association between eGFR and plasma proANP levels in the MetaCardis participants, consistent with previous reports[39]. These observations suggest that complimentary mechanisms connect variations in blood pressure regulation and glucose metabolism with kidney-heart function, which possibly enabled the identification of its microbial correlates in our study.

Our work adds a gut microbial component to this kidney-heart crosstalk. Indeed, the gut microbiome has been implicated in the production of metabolites affecting kidney function[12] (i.e., uraemic toxins), whereas impaired kidney function-related accumulation of such plasma metabolites and other nitrogenous waste has in-turn been proposed to induce gut dysbiosis[11]. We show that 1) some gut microbial plasma metabolites are indeed key mediators of the microbiome-eGFR associations, and 2) both these mechanisms potentially operate in parallel with microbial metabolites affecting kidney function (i.e., phenylacetylcarnitine and cinnamoylglycine lowering eGFR), and kidney function affecting the gut microbiome (i.e., eGFR regulating the

circulating levels of 4-cresyl sulphate, phenylacetylglutamine as key uraemic toxins and novel players like vanillactate and 3-(4-hydroxyphenyl)lactate). Based on our observations we posit that, under certain physiological conditions, the gut microbiome overproduces key metabolites that negatively impact the kidneys, tipping the balance towards impairment of its function. This kidney dysfunction, in turn, leads to accumulation of uraemic toxins that affect other organs, such as the heart and the brain[52], triggering a cascade of pathological changes. These early, subtle shifts can perpetuate and gradually intensify over time, contributing to physiological responses such as elevated arterial pressure with sustained proANP release, which may further compromise kidney function, creating a vicious cycle that besides kidney disease also accelerates CMD progression.

Dietary patterns involving high intake of processed meat plays a critical role in shaping the gut microbiome[56], with significant implications for kidney and cardiovascular health. Our findings imply that diet-induced shifts in the gut microbiome, such as the plausible upregulation of proteolytic fermentation, are linked to a lower eGFR and accumulation of microbial plasma metabolites including acetylcarnitine and cinnamoylglycine[57], which may contribute to both disease initiation and progression. Given these associations, dietary interventions targeting gut microbiome modulation could serve as a key strategy for preventing or delaying the onset of impairment of kidney function, CKD and CMD.

Recent genome-wide associations studies further link human genetic variation to the abundances as well as the genetic diversity of multiple microbial taxa of relevance to the cardiorenal axis and key metabolites identified in the current study (e.g. *Rothia*[58], *Ruminococcus*[59], *Bifidobacterium*[59–61] and *Faecalibacterium*[62,63]), suggesting that factors beyond diet can influence host-microbiome interactions to shape cardiorenal physiology and disease progression. Unravelling these factors could offer an opportunity to target key host-microbiome interactions, even those under host genetic regulation, to modify disease progression or the outcome.

Altogether, our study offers valuable insights into the potential gut microbial drivers of cardiorenal pathophysiology. However, several limitations remain: 1) our discovery cohort MetaCardis is a cross-sectional, case-control study mimicking the natural history of heart disease with a relatively modest sample size; 2) the coverage of our microbial taxa or functional annotation based on reference databases cannot be considered an exhaustive representation of human microbiome and has its own biases; 3) absence of metagenomic data in CLSA did not allow external replication of gut microbial taxonomic and functional signatures, which, unlike CLSA metabolomic profiles, would need to be validated in additional populations to ensure their generalizability; 4) the MetaCardis and CLSA participants are predominantly of European ancestry, hence further work in diverse populations is needed to ensure transferability of our findings to other ethnicities; 5) although we have comprehensively screened and adjusted for possible confounders, we cannot rule out residual confounding by unmeasured variables in our study; 6) the mediation and multivariate approaches used in the study do not account for the multicollinearity in the omics datasets; 7) we used the MR approach to add genetic evidence from completely unrelated populations to MetaCardis and CLSA, however it has inherent limitations[43], including the assumptions underlying instrument selection and horizontal pleiotropy. Lastly, we acknowledge that causality has not been determined in the current work and any implications of a causal role of the gut microbiome and its metabolites in cardiorenal physiology would need further investigation using experimental approaches.

In conclusion, our results demonstrate that human gut microbiome underpins early-stage variations in kidney-heart physiology within the normal kidney function range. Considering that impairment of kidney function often remains asymptomatic until very late stages

of the disease[64], these microbial contributors highlight the potential to bridge the current gap in our ability to diagnose or monitor the condition contributing to its growing prevalence and related disease burden globally[53]. This is important as 30% of people with T2D will likely develop CKD[53]; our work is timely and opens perspectives for identification of people at increased risk of deteriorating kidney function and adverse cardiovascular outcomes, thereby aiding precision medicine efforts targeted at reducing the global burden of CMDs.

## Methods

### Ethics statement

The MetaCardis study received ethical approval from CPP Ile-de-France, the University of Leipzig Ethics Committee, the Office for Human Research Protections (OHRP), and the Capital Region of Denmark Ethics Committees, and is registered on ClinicalTrials.gov (NCT02059538). The goal of the trial was to understand how the qualitative and quantitative changes in the gut microbiome contribute to the pathogenesis of CMDs and their associated co-morbidities. The trial additionally aimed to uncover novel microbial signatures that could help diagnose and/or predict the natural evolution of CMDs enabling future personalized medicine. All participants provided written informed consent, and clinical investigations were conducted in accordance with the Declaration of Helsinki II. The MetaCardis data analyses presented in the current study are covered under these approvals and no separate ethics review was required.

Ethical approval for "the EGEA study" study was obtained from the relevant institutional review board committees (Cochin Port-Royal Hospital and Necker-Enfants Malades Hospital, Paris: n° 01-07-07, 04-05-03, 04-11-13 and 04-11-18). Written informed consent was obtained from all adult participants and child's legal guardians at both surveys. The EGEA data analyses presented in the current study are covered under this approval and no separate ethics review was required.

Detailed description of the recruitment process, data availability and bio-clinical phenotyping for the CLSA is covered here[44]. The CLSA governance structure including the ethics oversight are further detailed on www.clsa-elcv.ca. Additionally, Imperial Research Ethics Committee approved the usage of CLSA data for the analyses used in the current manuscript under the application number 21IC7388.

### MetaCardis population

The European MetaCardis project included 275 healthy controls and 1602 individuals at various stages of dysmetabolism and IHD, aged 18–76 years, 49.4% males and 50.6% females, recruited from France, Denmark, and Germany between 2013 and 2015. Sex was included as part of the study design, with each participant reporting their sex at enrolment, which was recorded by the clinical investigators in the electronic case report form. In France, healthy participants were selected from a subgroup of individuals who had undergone clinical and bio-clinical phenotyping as part of the Nutrinet Study[65], and they were examined at the clinical investigation Centre (CIC-Paris Est) at Pitié-Salpêtrière, Paris. In Germany, healthy individuals were recruited through open advertisements at the University Hospital in Leipzig, in local newspapers, and via the website of the Integrated Research and Treatment Centre for Adiposity Diseases (IFB). In Denmark, participants were drawn from the 'Health 2006 study' conducted at the Research Centre for Prevention and Health at Glostrup University Hospital and were recalled for this study. Exclusion criteria included known factors influencing the gut microbiome, such as antibiotic use within the previous three months, a history of abdominal cancer (with or without radiation therapy), intestinal resection (except appendectomy), inflammatory or infectious diseases (including Hepatitis B/C and HIV). Additionally, individuals with a history of organ transplantation, those on immunosuppressive therapy, with severe kidney failure (eGFR

<50 ml/min/1.73 m²), drug or alcohol dependency, pregnancy, or breastfeeding were excluded. Participants were recruited following a telephone interview and a medical history review where available, as well as during hospital clinical consultation. The promoter of the study was Assistance Publique-Hopitaux de Paris (APHP).

### EGEA population

The EGEA was designed to identify the genetic and environmental factors and their interactions involved in asthma and asthma-related phenotypes[48]. It combines an initial group of asthma cases with their first-degree relatives, and a group of population-based participants with three surveys over 20 years (EGEA1: 1991-1995, EGEA2: 2003-2007 and EGEA3: 2011-2013). The whole study population (N = 2,120) included 388 asthmatic probands recruited in chest clinics of five French cities (Lyon, Grenoble, Marseille, Montpellier, Paris) and their 1,317 first-degree relatives plus 415 population-based controls. EGEA subjects are of European ancestry and were born in France. Data collected through standardized questionnaires and clinical examination included extensive phenotypic characterization mainly related to asthma and allergy and data on environmental exposures and lifestyle factors (see https://cohorte-egea.fr/en for details).

A total of 1351 EGEA adults examined at the second survey (EGEA2) with plasma samples available for GCMS based metabolic profiling were included in this study. After applying GCMS quality control procedures, there were 1298 subjects with 4-cresol data of which 1,194 also had SNP data for the genome-wide association analysis. 51% of these individuals were females and on average 42.9 years old. Additionally, 41.5% of these individuals had history of asthma, 28.7% were living with overweight and 10% with obesity, while 3.8% and 5.8% reported having diabetes or heart disease, respectively.

### CLSA population

The CLSA study follows 50,000 Canadian individuals to understand and address the needs of an aging population. The cohort of 51,338 participants, aged 45–85 years at enrolment, were recruited from 2010-2015 and are being tracked longitudinally every 3 years. Only individuals with metabolomics data availability in the CLSA study were included in the current study. Additionally, our analyses were restricted to individuals of European ancestry (n = 8669 for disease incidence and n = 9135 for mortality) as only unrelated European ancestry individuals were used for genome-wide association analyses of the metabolites by Chen et al.[45] based on the genomic data in CLSA[66].

### MetaCardis: bio-clinical and lifestyle phenotyping

Clinical measurements were made using standardized operating procedures concluded prior to patient recruitment in the MetaCardis study. Information on anthropometrics, lifestyle and bioclinical variables such as age, sex, BMI, food intake, smoking status, physical activity, socio-economic factors, physiological parameters and medication was collected.

Habitual dietary information was obtained using food-frequency questionnaires adapted to the cultural habits of each of country of recruitment, relative validity of which has been assessed previously[67]. Smoking status was obtained from a standardized questionnaire while information on physical activity was assessed using the Recent Physical Activity Questionnaire. Drug intake was assessed by either recall or from medication list, and a medical specialist interviewed study participants about adherence to prescribed medication.

Disease definitions followed international criteria, with obesity defined according to WHO criteria, metabolic syndrome according to the International Diabetes Federation[68] and prediabetes and T2D by the American Diabetes Association[69] and hypertension according to the American College of Cardiology and American Heart Association[70]. IHD and HF were defined according to the American College of

Cardiology, American Heart Association and the Heart Failure Society of America. Specifically, overweight and obesity were defined as BMI < = 25 kg/m$^2$ and obesity > 30 kg/m$^2$, respectively. Prediabetes was defined by fasting plasma glucose ≥ 5.6 and <6.9 mmol/l and/or HbA1c ≥ 5.7 and < 6.4% whereas TD2 was defined as fasting plasma glucose ≥ 7 mmol/l and/or HbA1c ≥ 6.5% and/or subjects taking any glucose lowering agents. Hypertension was defined as systolic blood pressure > 140 mmHg and/or diastolic blood pressure > 90 mmHg and/or subjects taking anti-hypertensive drugs. Subjects with IHD were defined as having a history of coronary heart disease with an acute event (>15 days) or chronic disease (diagnosed by angiography angiogram) with normal heart function, and documented (HF) as demonstrated by echocardiography-evaluated (LVEF) < 45%. Renal function was assessed via eGFR calculated using the Modification of Diet in Renal Disease (MDRD) equation[71].

Blood was sampled in the morning after an overnight fast. Plasma and serum samples were stored at the clinical centres at -80 °C until shipment to central laboratory facility. Fasting plasma glucose, total and HDL cholesterol, triglycerides, creatinine and HbA1c levels were measured using enzymatic methods. LDL-cholesterol concentrations were measured enzymatically for German participants, and values for French and Danish study participants were calculated based on the Friedwald equation. Liver enzymes, i.e., alanine aminotransferase, aspartate aminotransferase, and gamma-glutamyl transferase levels were measured using enzyme-coupled kinetic assays. Plasma pro-ANP was measured using processing-independent assay[72].

### MetaCardis: gut microbiome analyses

**Stool sample collection.** Stool samples were collected and processed according to the International Human Microbiome Standards (IHMS) guidelines (SOP03V1). Briefly, samples were collected by patients at home and immediately stored at 20 °C until they were transported on dry ice and frozen 4 to 24 h later at −80 °C in plastic tubes at the biobanks of corresponding recruitment centres. While the exact time of stool collection (e.g., morning vs. other times) was not captured, participants were instructed to provide the first stool sample of the day of investigation as part of the MetaCardis protocol.

**Stool sample processing and metagenomic analyses.** The metagenomic analyses for the entire MetaCardis study was conducted using standardized protocols at a single site (Metagenopolis, Paris) over a period of 18 months. No significant bias of the sequencing date for different Metacardis groups was observed as reported in our previous study[6]. The detailed methodology has also been described previously[6]. Briefly, however, total faecal DNA was extracted following the IHMS guidelines (SOP-07V2 H) and samples were sequenced using ion-proton technology resulting in 23.3 ± 4.0 million (mean ± SD) single-end short reads of 150-base on average. Low quality nucleotides, any remaining sequencing adapters, human and other possible food contaminant DNA were filtered next, followed by mapping these high-quality reads against the 9.9 million integrated gene catalogue (IGC) of the human microbiome[73]. A two-step process was next employed to derive the gene abundance profiling: 1) the unique reads (i.e., reads mapped to a unique gene in the catalogue) were attributed to their corresponding genes, whereas 2) the shared reads (i.e., reads mapping to multiple genes in the catalogue with same alignment score) were attributed according to the ratio of their unique mapping counts. To reduce any technical bias due to different sequencing depths, the gene abundance profiles were next rarefied to 10 million reads per sample by random sampling of 10 million mapped reads without replacement. The resulting rarefied gene abundance table was then normalized using the FPKM approach, which accounts for gene length and total mapped read counts.

Metagenomic species are defined as co-abundant gene groups with more than 500 genes corresponding to a microbial species,

n = 1,436, as described previously[17,21]. Additionally, an metagenomic Operational Taxonomic Unit (mOTU) approach was used to quantify microbial taxa as described in detail previously[17,74], whereas microbial gene richness or gene count was derived by counting the number of genes that were detected at least once in each sample, using the average number of genes counted in ten independent rarefaction experiments as reported previously[6,17]. The functional modules (such as KEGG pathways) were quantified by mapping metagenome reads to the IGC gene catalogue post rarefaction followed by binning for functional categories, as described previously[17]. The mOTU based taxa, MGS abundances and functional modules were corrected for bacterial cell count to derive quantitative microbiome profiling as described previously[6,17,18].

**Customized functional module analysis.** Customized functional module sets included GMMs covering bacterial and archaeal metabolism specific to the human gut environment with a focus on anaerobic fermentation processes, which have been described in detail previously[6,17,18]. GMM abundances were also corrected for bacterial cell count.

### MetaCardis: metabolite profiling

A comprehensive metabolic phenotyping strategy combining in-house analysis by proton nuclear magnetic resonance ($^1$H-NMR) spectroscopy, gas chromatography coupled to mass spectrometry (GC-MS) and targeted ultra-performance liquid chromatography coupled to tandem mass spectrometry (UPLC-MS/MS) with untargeted UPLC-MS data generated by Metabolon were employed in the MetaCardis study, as described in detail previously[6]. Briefly, $^1$H-NMR experiments were carried out using a Bruker Avance spectrometer (Bruker GmbH). Structural assignment was performed using data from literature, HMDB, S-Base (Bruker GmbH) and in-house databases[75]. $^1$H-NMR spectra were pre-processed and exported to Matlab as previously reported[76], followed by absolute metabolite quantification using Bruker's——In Vitro Diagnostics for research (IVDr) quantification algorithms B.I.LISA, B.I.QUANT PS and B.I.QUANT UR. For semi-targeted profiling using GC-MS described in detail previously[6], briefly, serum samples were subjected to methanol protein precipitation, evaporated to dryness, derivatized and injected to an Agilent 7890B-5977B Inert Plus GC-MS system. Peaks were annotated using Fiehn library (Agilent G1676AA Fiehn GC/MS Metabolomics RTL Library, User Guide, Agilent Technologies, https://www.agilent.com/cs/library/usermanuals/Public/G1676-90001_Fiehn.pdf). Quality Control (QC) samples were derived from pooling equal amounts of all serum sample in the study. Study samples were randomized and prepared in batches of sixty (60), along with 6 QC samples and 2 blank (100 µL of H$_2$O instead of serum) samples. Every preparation batch was injected over 48 h, with half of the samples (30 study, 3 QC and 1 blank samples) analyzed right after preparation and the rest after 24 h of storage at -20oC. The absolute quantification of key methylamines and carnitines was carried out by spiking samples with internal standards followed by in house UPLC-MS/MS also described in detail previously[6]. Serum samples were also profiled by Metabolon (Durham, NC) using a UPLC-MS based methodology[77]. Annotations were performed by comparing sample features with ion features in a reference database of pure chemical standards and previously detected unknowns (denoted X-00000), followed by detailed visual inspection and quality control as reported[78].

The sample preparation order was randomized across the MetaCardis study such that each sample preparation batch included samples from all study groups. For MS untargeted assays, median batch-correction was performed by adjusting batch-wise study sample variable medians according to a scalar derived from adjusting pooled reference sample medians such that pooled reference sample medians remained identical across all batches. Finally, when duplicate

metabolites were quantified from multiple methodologies, we prioritized measurements based on the analytical quality of the data as per the criteria described previously[6].

**Plasma GC-MS analysis of 4-cresol in EGEA.** Semi-quantitative analysis of 4-cresol in EGEA was carried out in the Shimadzu metabolomic platform at Kyoto University using previously described GC-MS methods. Briefly, 50 μL plasma aliquots were mixed with a solution of the internal standard 2-isopropylmalic acid and extraction solvent. Following centrifugation, the supernatant was mixed with a solution of methoxyamine and N-methyl-N-trimethylsilyltrifluoroacetamide (GL science, Tokyo, Japan). GC-MS analysis was performed using a GCMS-QP2010 Ultra (Shimadzu, Kyoto, Japan). The derivatized metabolites were separated on a DB-5 column (Agilent Technologies, Palo Alto, CA). Chromatographic peak of 4-cresol was identified by comparing its mass spectral pattern to that in the NIST library and Shimadzu GC-MS Metabolite Database v1, followed by confirmation through comparison of retention index between samples and the corresponding authentic standard.

The 1,351 EGEA plasma samples were run by batch (N = 34 batches; 40 samples by batch on average). During the experiment, there was a maintenance of the GCMS instrument, which led us to analyze the pre-maintenance and post-maintenance datasets separately. After removing a few outliers, the pre-maintenance (set A) included 724 samples while the post-maintenance set (set B) included 574 samples. To minimize variation due to noise and normalize the data, we used a batch correction method based on smoothing spline regression of metabolite signal on order of injection within batch. We then applied an inverse normal transformation to the batch-corrected 4-cresol measurement.

### Statistical analysis

The sample sizes ranged from 200–275 for the metabolically healthy individuals, and 1462–1602 for individuals with cardiometabolic diseases in the MetaCardis population. This is due to missing data for certain questionnaires, typically socioeconomic or lifestyle related. We did not impute clinical or lifestyle-related data and have presented the sample sizes per analyses in the respective Figure/ Data legends and Source data files.

**Observational association analyses.** Univariate analyses included univariable Analysis of Variance Using Distance Matrices (ADONIS) (R package vegan, v2.6-8), Spearman's rank-based correlation coefficients (R package ppcor v1.1), Wilcoxon rank-sum test, one-way Kruskal-Wallis with Dunn-Bonferonni *post hoc* testing (R package FSA, v0.10) and multiple linear regression including analysis of covariance (ANCOVA) using rank-normalized data and Cox-proportional hazards regression using either right-censoring (for mortality) or interval-censoring (for disease incidence) as appropriate using R packages survival and IcenReg, v2.0.16, respectively. Statistical tests were two-sided, P-values were corrected for multiple testing using Benjamini-Hochberg (BH) false discovery rate (FDR) criterion, and FDR < 0.1 was considered significant unless otherwise specified. Participants with missing data were excluded based on the variables considered in each analysis separately unless otherwise specified.

Multivariate analyses included multivariable ADONIS with 20,000 permutations using vegan, v2.6-8 package in R.

For MetaCardis study, microbiome ecological, genus and MGS data were used without any transformation for Spearman and partial Spearman correlation analyses, whereas microbiome function data (i.e., GMM and KEGG pathways), metabolites and clinical data were ranknormalized (RNOmni, v1.01.2) prior to any linear modelling. Specific details for each analysis are included in the respective Figure and Data legend. In the CLSA study, metabolites levels were transformed using the natural logarithm with values above or below mean ± s.d. replaced by the respective upper or lower bound. Metabolites levels were then rescaled to have a mean of zero and s.d. of one in alignment with the analyses in EPIC-Norfolk data[35]. The eGFR data in CLSA was winsorized at 15 and 200 ml min$^{-1}$ per 1.73 m$^2$ in alignment with CKDGen[46].

**Observational predictive analyses.** We employed multivariate CA-PLS analyses using five-fold cross-validations and 1000 random Monte Carlo iterations as in[26] using MATLAB (v21.2) and multivariable Cox-proportional hazards models using either right-censoring (for mortality) or interval-censoring (for disease incidence) as appropriate with five-fold cross-validations and 1000 permutations using R packages survival and IcenReg, v2.0.16, respectively.

**Observational Mediation analyses.** We assumed linear dependency among key relationships of microbial features-metabolites and cardio-renal variables (i.e., eGFR and proANP) when we tested the role of metabolites as mediators of these inter-relationships using the mediation, v4.5.0 package in R. Multiple testing correction per category of microbiome features using BH criteria was applied with FDR < 0.1 considered significant.

**Causal inference analyses using Mendelian randomization.** MR is a genetic epidemiology tool that uses genetic variants, i.e., single nucleotide polymorphisms (SNPs) associated with an exposure as instrumental variables for inferring the association of the genetically predicted levels of an exposure with an outcome. The method involves three basic steps, i.e., 1) identification of instruments, 2) derivation of genetic associations for instruments with the exposure and outcome, and 3) MR analyses followed by sensitivity analyses to test the validity of the MR findings.

**Instrument selection.** Instruments must be selected to meet three core assumptions of MR: 1) the genetic instrument is associated with the exposure (relevance assumption); 2) there are no measured or unmeasured confounders of the instrument and outcome (independence assumption); and 3) there is no independent pathway between the instrument and the outcome other than through the exposure (exclusion-restriction assumption). To meet the relevance assumption, a genome-wide significant cutoff of $P <= 5.0e-08$ is generally prescribed[79]. However, this is not always feasible especially when working with -omics variables like the metabolites, proteins and gut microbiome variables. Multiple studies[59,60,80-82] have previously used P-value thresholds ranging from $P <= 1.0e-05$ to $2.5e-08$ to find genetic instruments of the gut microbiome variables with the objectives of maximizing explained variance[81] and inclusion of enough SNPs to be able to perform sensitivity analyses[61]. We ensure that genetic variants are still strong instruments by calculating the F-statistic and excluding any genetic variant with F-statistics <10. We chose $P <= 1.0e-05$ for finding instruments for our microbial- and host-co-metabolites and NPPA whereas $P <= 5.0e-08$ was used for finding instruments for eGFR as exposure. SNPs were matched for their availability in the outcome GWAS followed by removal of SNPs exhibiting minor allele frequency (MAF) < 1% (except for 4-cresol where we used MAF < 10% owing to EGEA sample size), ambiguous and palindromic SNPs exhibiting MAF > 0.42 and those within ± 250 kb of the MHC region. Remaining SNPs were then subjected to clumping procedure employing linkage disequilibrium threshold of $r^2 < 0.001$ in a 10MB window using a European reference population using ieugwasr v1.01 R package.

Genetic instruments were derived from our in house GWAS analysis in the EGEA cohort for 4-cresol (Supplementary Data 16), whereas publicly available GWAS summary statistics were used for all other traits including microbial- and host co-metabolites[45], NPPA[47], eGFR[46].

**Genome-wide association analyses for 4-cresol in the EGEA cohort.** Genotyping of EGEA subjects was done using the Illumina 610-Quad array at the Centre National de Génotypage (CNG, Evry, France), as part of the European GABRIEL asthma consortium[83] Quality control procedures have been described in detail elsewhere[83]. A total of 531,401 autosomal SNPs remained after QC and were available in 667 and 527 individuals with 4-Cresol data from sets A and B respectively. Genotype imputation was performed through the Michigan imputation server (https://imputationserver.sph.umich.edu/index.html#!) using the Minimac4 software[84] and the 1000 Genomes Phase 3-version 5 reference panel. For analysis, we kept bi-allellic SNPs with imputation quality score (rsq) ≥ 0.5 and minor allele frequency ≥ 1%, which corresponded to 7.77 million SNPS for both datasets A and B.

The genome-wide association analysis between 4-cresol and individual SNPs was based on a linear mixed model assuming an additive genetic model for the SNP effect and adjusting for age and sex. We used the GEMMA software[85], which allows for accounting familial relationships through a relatedness matrix. The estimates of SNP effect from the two sets A and B were combined using a fixed-effect meta-analysis with inverse variance weighting implemented in Stata V14.1. The test of the combined SNP effect on 4-cresol was based on a Wald test. We observed little evidence of inflation in this test statistic (genomic inflation factor l = 1.036).

**Univariable MR analysis.** We performed two-sample MR analyses using the IVW multiplicative random effects as the main method which operates under the assumption of balanced pleiotropy[86] and derives its MR effect estimate from the meta-analysis of the individual SNP effects (*i.e.* (Wald's) ratio of the SNP effect on outcome by the SNP effect on the exposure) weighted by the inverse of their variance (*i.e.* squared ratio of the SNP standard deviation on the outcome by the SNP standard deviation on the exposure). In each case, the allele exhibiting positive association with the exposure was set as the effect allele.

One of the key sources of potential bias in the MR approach is the violation of exclusion-restriction assumption or phenomenon of horizontal pleiotropy[87]. We employed multiple methods like Weighted Median[88], MR Egger[89], contamination mixture[90] and MR-PRESSO[91], which make varied assumptions about the validity of the instruments. Moreover, we applied outlier-correction using Q-based statistics[92] when nominally significant excess heterogeneity as measured by the Q-statistic was detected. We additionally used scatter plots and funnel plots to visually assess the SNP-exposure and SNP-outcome relationships for inferring pleiotropy or assessing the impact of outlier correction using Mendelian Randomization, v0.10, package in R.

Our initial analysis included IVW with multiple testing correction using BH criteria where FDR < 0.1 was considered significant. Cochrane's Q, P < 0.05 or I² > 25% was taken as evidence of heterogeneity. We used both MR-PRESSO and Q-statistics to identify outlying SNPs and looked for evidence of significance in Weighted Median, MR-Egger and contamination mixture-based estimates pre- and post-Q-based outlier correction.

## Data availability
Raw shotgun sequencing data generated in this study have been deposited in the European Nucleotide Archive under accession codes PRJEB41311, PRJEB38742 and PRJEB37249 with public access. The Serum NMR metabolome data generated in this study have been deposited to Metabolights with accession number "MTBLS3429", and can additionally be requested by contacting the corresponding authors. The Serum GC-MS and isotopically quantified serum metabolites (UPLC–MS/MS) data generated in this study have been deposited in MassIVE database with accession numbers "MSV000088042 [https://doi.org/10.25345/C5CV76]" and

"MSV000088043 [https://doi.org/10.25345/C58246]", respectively. In adherence to EU and national privacy laws, unrestricted access to individual phenotypic data cannot be provided for the MetaCardis study. Interested researchers, wishing to access individual phenotypic data would need to submit argued applications to the relevant National Data Protection Agencies. These are the Danish Data Protection Agency (https://www.datatilsynet.dk/english) for phenotypic data from study participants recruited in Denmark, the Federal Commissioner for Data Protection (https://www.bfdi.bund.de/EN/Home/home_node.html) for phenotypic data from study participants recruited in Germany and the Commission Nationale Informatique & Libertés (https://www.cnil.fr/en/home) for phenotypic data of study participants recruited in France. Application procedures are given on the outlined websites. If such permission is granted, phenotypic data will be then made available by the corresponding authors within 5 weeks. All omics and phenotypic data from the Canadian Longitudinal Study on Aging (www.clsa-elcv.ca) are protected by Canadian personal data privacy laws. The CLSA data are only available to researchers who meet the criteria for access to de-identified CLSA data. Source data are provided with this paper.

## Code availability
No custom code or algorithm was used for the analyses conducted in this work.

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

## Acknowledgements

This study was initiated and funded by the European Community's Seventh Framework Programme (FP7/2007-2013): MetaCardis, grant agreement HEALTH-F4-2012-305312, a Joint Programming Initiative (A healthy diet for a healthy life; 2017-01996_3), a Transatlantic Networks of Excellence Award from the Leducq Foundation (17CVD01) and by the NIHR Imperial Biomedical Research Centre. GC-MS analysis in EGEA was funded by a grant from the Agence Nationale pour la Recherche (METABASTHMA, ANR-17-CE14-0042-01). Genotyping in EGEA was supported by grants from the European Commission (No. LSHB-CT-2006-018996-GABRIEL) and the Wellcome Trust (WT084703MA). We thank the EGEA study participants, and Hamida Mohamdi and Patricia Margaritte-Jeannin for their contribution to this work. This work was also supported by the Agence Nationale de la Recherche (ANR) with the MetaGenoPolis grant (ANR-11-DPBS-0001). Infrastructure support for this research was provided by the NIHR Imperial BRC. This research is also funded by grants from the French National Research Agency (ANR-10-LABX-46 [European Genomics Institute for Diabetes]), from the National Centre for Precision Diabetic Medicine – PreciDIAB, which is jointly supported by the French National Agency for Research (ANR-18-IBHU-0001), by the European Union (FEDER), by the Hauts-de-France Regional Council (Agreement 20001891/NP0025517), by the European Metropolis of Lille (MEL, Agreement 2019_ESR_11) and by Isite ULNE (R-002-20-TALENT-DUMAS), also jointly funded by ANR (ANR-16-IDEX-0004-ULNE), the Hauts-de-France Regional Council (20002845) and by the European Metropolis of Lille (MEL) and ERC Generator Grant "Richness" (R-ERCGEN-23-003-DUMAS) from the University of Lille. This research was also conducted as part of the CNRS–Imperial-ULille International Research Project in Integrative Metabolism "METABO-LIC". The authors also acknowledge that this research was made possible using the data/biospecimens collected by the Canadian Longitudinal Study on Aging (CLSA). Funding for CLSA is provided by the Government of Canada through the Canadian Institutes of Health Research (CIHR) under grant reference: LSA 94473 and the Canada Foundation for Innovation, as well as the following provinces, Newfoundland, Nova

Scotia, Quebec, Ontario, Manitoba, Alberta, and British Columbia. This research has been conducted using the CLSA metabolomics data version 1, CLSA Comprehensive baseline dataset (v7), Comprehensive follow-up 1 dataset (v5), CLSA participant status data under Application Number 2104039. The CLSA is led by Drs. Parminder Raina, Christina Wolfson and Susan Kirkland. K.Che. is supported by the NIHR Imperial Biomedical Research Centre (BRC) through a fellowship jointly funded by the Cardiovascular and Multimorbidity Themes. K.Che also acknowledges the support of the Medical Research Council Skills Development Fellowship (grant no. MR/S020039/1) and the Wellcome Trust-funded Institutional Strategic Support Springboard Fellowship (grant no. 204834/Z/16/Z). R.C. is the recipient of the Walter Benjamin Fellowship from the German Research Association (DFG) project number 462524713 and the EASO-Novo Nordisk Foundation New Investigator Award: Clinical Research, project number NNF25SA010378. L.H. was a recipient of an MRC Intermediate Research Fellowship in Data Science (grant number MR/L01632X/1, UK Med-Bio) and is supported by the European Union's Horizon 2020 research and innovation programme (grant agreement number 874583). A.R.M. was recipient of a Doctoral Training Centre PhD scholarship (MR/K501281/1), Imperial College PhD-scholarship (EP/M506345/1) and a La Caixa studentship. A.L.N. received a Portuguese Foundation for Science and Technology (SFRH/BD/52036/2012) scholarship. F.M. and D.G. are recipients of the INSERM International Research Project DIABETOMARKERS. P.A. is the recipient of a Career Development Award from the Medical Research Council (Grant No. MR/Y010051/1). V.Z. acknowledges funding support from the United Kingdom Research and Innovation Medical Research Council grant MR/W029790/1 and the UK Dementia Research Institute, which receives its funding from UK DRI Ltd, funded by the UK MRC, Alzheimer's Society and Alzheimer's Research UK. I.T. acknowledges support from the Imperial College British Heart Foundation Centre for Research Excellence (RE/24/130023) and the NIHR Imperial Biomedical Research Centre. S.K.F. acknowledges funding support from EU: IMMEDIATE consortium, DFG: SFB1470, TRR412 and EXC3118 (ImmunoPreCept), and from DZHK (German Centre for Cardiovascular Research). M.S. acknowledges grant support from the Deutsche Forschungsgemeinschaft (DFG), EXC3105-1. K.Cle. also acknowledges support from the CNIEL (Centre National Interprofessional de l'Economie Laitiere) and BNP-Cardiff for grant support on nutritional aspects in this cohort, the Inserm (IRP programme), the ANR (NutrimCheck project) and Horizon Europe, European Commission EIC Pathfinder "Nutrimune", European community. M.-E.D. acknowledges funding support from the EU IMMEDIATE consortium under contract number 101095540 and UKRI Innovate UK under contract number 101095556 and by the National Institute for Health Research (NIHR) Imperial Biomedical Research Centre, as well as grants from Guts UK (DG201808), Diabetes UK (19/0006059), and a Medical Research Council grant to M.-E.D. and P.F. (MR/X010155/1). The Novo Nordisk Foundation Centre for Basic Metabolic Research is an independent research institution at Faculty of Health and Medical Sciences, the University of Copenhagen, partially funded by an unrestricted donation from the Novo Nordisk Foundation (NNF23SA0084103). The opinions expressed in this manuscript are the author's own and do not reflect the views of the Canadian Longitudinal Study on Aging. Also, despite being funded by the European Union, views and opinions expressed are those of the author(s) only and do not necessarily reflect those of the European Union or European Health and Digital Executive Agency (HADEA). Neither the European Union nor the granting authority can be held responsible for them.

## Author contributions

K.Che., M.-E.D., K.Cle, S.D.E., and O.P. developed the present study concept and protocol. K.Cle (Coordinator and principal investigator), M.-E.D., S.D.E., O.P., P.B., M.S., J.R., J.B.N., D.G. and F.B. conceived the study design of the MetaCardis consortium. MetaCardis cohort recruitment, phenotyping and lifestyle recording were conducted by J.A.-W., T.N., R.C., C.L., L.K., T.H., T.H.H., H.V., N.B.S., H.K.P., J.N., S.H., M.Blu. Meta-Cardis consortium data curation was undertaken by R.C., S.A., S.K.F., J.A.-W., and T.N. Faecal microbial DNA extraction and shotgun sequencing N.P., E.L.C., S.F., H.R., B.Q., N.G., M.Ber., P.B.L., K.D.S., P.G., J.D.Z., I.L., J.M.O., P.F. Bacterial cell count measurement: G.F., SVS. Serum and urine metabolome profiling (MetaCardis): L.H., J.C., A.Myr, D.G., F.M. MetaCardis metabolite annotation by J.C., A.Myr, M.O., A.L.N. Pro-ANP measurements by PDM and J.-P.G. Bioinformatics and statistical analyses: K.Che, S.F., S.K.F., B.J., L.P.C., L.M.G., E.P., Ebel, F.P., P.A., F.P.C., R.P.T., I.C.D. GC-MS analysis and GWAS of 4-cresol in EGEA study: E.Bou, F.D., M.L., D.G., K.S., T.A.S. and F.M. CLSA data access and analysis: K.Che, M.J., AMan, P.R., M.L., M.-E.D. Mendelian Randomization: KChe with input from V.Z., A.D., I.T. The manuscript was drafted by KChe and M-ED with inputs from R.C., S.K.F., O.P., K.Cle and S.D.E. All authors approved the final version for publication.

## Competing interests

K.Cle. has held a collaborative research contract with Danone Research in the context of MetaCardis project. O.P. is a co-founder of GutCRINE. F.B. is shareholder of Implexion pharma AB and Roxbiosens, receives research grants from Biogaia AB and Novo Nordisk A/S and is on the scientific advisory board of Bactolife A/S. V.T. is shareholder of Roxbiosens. K.S. and T.A.S. are employees of Shimadzu, Kyoto, Japan. M.Blu. received honoraria as a consultant and speaker from Amgen, AstraZeneca, Bayer, Boehringer Ingelheim, Daiichi-Sankyo, Lilly, Novo Nordisk, Novartis, and Sanofi. The remaining authors declare no competing interests.

## Additional information

Kanta Chechi [1,2,3,62] ✉, Rima Chakaroun [4,5,6,62], Antonis Myridakis [1,62], Sofia K. Forslund-Startceva [7,8,9,10,62], Sebastien Fromentin [11,62], Trine Nielsen [12,13,14,15], Judith Aron-Wisneswky [16,17], Eugeni Belda [16,18], Edi Prifti [16,18], Pierre Bel Lassen [16,17], Gwen Falony [19,20,21,22], Sara Vieira-Silva [19,20,21,23,24], Julien Chilloux [1], Kazuhiro Sonomura [25], Lesley Hoyles [1,26], Laura Martinez-Gili [1], Francesco Pallotti [1,27], Petros Andrikopoulos [1], Francesc Puig-Castellví [28], Romina Pacheco Tapia [28], Inés Castro-Dionicio [28], Hugo Roume [11], Nicolas Pons [11], Emmanuelle Le Chatelier [11], Benoit Quinquis [11], Nathalie Galleron [11], Magali Berland [11], Michael T. Olanipekun [1], Manyi Jia [1], Angelos Manolias [1], Bridget Holmes [29,30], Solia Adriouch [16], Matthias Blüher [4,5], Luis Pedro Coelho [31], Kévin Da Silva [11], Pilar Galan [32], Boyang Ji [33,34], Ana Luisa Neves [1,35], Christine Rouault [16], Joe-Elie Salem [36], Valentina Tremaroli [6], Tue H. Hansen [12,13,14], Nadja B. Søndertoft [12], Christian Lewinter [12], Helle K. Pedersen [12], The MetaCardis Consortium*, Peter D. Mark [37], Jens P. Goetze [38], Lars Køber [38,39], Henrik Vestergaard [12,15], Torben Hansen [12,40], Jean-Daniel Zucker [16,18], Taka-Aki Sato [25], Serge Hercberg [32], Fredrik Bäckhed [6,41], Ivica Letunic [42,43], Jean-Michel Oppert [17], Jens Nielsen [33,34], Jeroen Raes [19,20], Ioanna Tzoulaki [3,44,45], Abbas Dehghan [3,45], Verena Zuber [3,45,46], Emmanuelle Bouzigon [47], Mark Lathrop [48], Parminder Raina [49,50], Philippe Froguel [28,51], Fumihiko Matsuda [52], Florence Demenais [47], Dominique Gauguier [48,52,53], Michael Stumvoll [4,5], Peer Bork [8,31,43,63], Oluf Pedersen [12,39,54] ✉, S. Dusko Ehrlich [11,55] ✉, Karine Clément [16,17] ✉ & Marc-Emmanuel Dumas [1,2,28,48] ✉

[1]Division of Systems Medicine, Department of Metabolism, Digestion and Reproduction, Faculty of Medicine, Imperial College London, London, United Kingdom. [2]Genomic and Environmental Medicine Section, National Heart and Lung Institute, Imperial College London, London, United Kingdom. [3]Department of Epidemiology and Biostatistics, School of Public Health, Faculty of Medicine, Imperial College London, London, United Kingdom. [4]Helmholtz Institute for Metabolic, Obesity and Vascular Research (HI-MAG) of the Helmholtz Zentrum München, University of Leipzig, Leipzig, Germany. [5]Medical Department III – Endocrinology, Nephrology, Rheumatology, University of Leipzig Medical Center, Leipzig, Germany. [6]The Wallenberg Laboratory, Department of Molecular and Clinical Medicine, Institute of Medicine, Sahlgrenska Academy, University of Gothenburg, Gothenburg, Sweden. [7]Experimental and Clinical Research Center, Charité–Universitätsmedizin & Max-Delbrück Center, Berlin, Germany. [8]Max Delbrück Center for Molecular Medicine (MDC), Berlin, Germany. [9]Charité University Hospital, Berlin, Germany. [10]DZHK (German Centre for Cardiovascular Research), Partner Site Berlin, Berlin, Germany. [11]Université Paris-Saclay, INRAE, MGP, Jouy-en-Josas, France. [12]Novo Nordisk Foundation Center for Basic Metabolic Research, Faculty of Health and Medical Sciences, University of Copenhagen, Copenhagen, Denmark. [13]Department of Clinical Medicine, Faculty of Health and Medical Sciences, University of Copenhagen, Copenhagen, Denmark. [14]Medical Department, Zealand University Hospital, Køge, Denmark. [15]Steno Diabetes Center Copenhagen, Copenhagen, Denmark. [16]Sorbonne Université, Inserm, Nutrition and Obesities: Systemic Approach Research Group, Paris, France. [17]Assistance Publique–Hôpitaux de Paris, Pitié-Salpêtrière Hospital, Nutrition Department, Paris, France. [18]Sorbonne Université, IRD, UMMISCO, Paris, France. [19]Laboratory of Molecular Bacteriology, KU Leuven, Leuven, Belgium. [20]Center for Microbiology, VIB, Leuven, Belgium. [21]Institute of Medical Microbiology and Hygiene, University Medical Centre Mainz, Mainz, Germany. [22]Host-Microbe Interactomics Group, Wageningen University & Research, Wageningen, Netherlands. [23]Systems Biology and Multiomics Research Group, IREC, UCLouvain, Brussels, Belgium. [24]Institute of Molecular Biology (IMB), Mainz, Germany. [25]Life Science Research Center, Technology Research Laboratory, Shimadzu Corporation, Kyoto, Japan. [26]Department of Biosciences, Nottingham Trent University, Nottingham, United Kingdom. [27]Department of Medicine and Surgery, University of Enna "Kore", Enna, Italy. [28]METAB-OMICS UMR8199/1283 CNRS, INSERM, Institut Pasteur de Lille, Lille University Hospital, University of Lille, Lille, France. [29]Global Nutrition Department, Danone Research, Palaiseau, France. [30]Food and Agriculture Organization of the United Nations (FAO), Rome, Italy. [31]Structural and Computational Biology, European Molecular Biology Laboratory, Heidelberg, Germany. [32]Nutritional Epidemiology Research Team (EREN), CRESS, Inserm, Bobigny, France. [33]Department of Life Sciences, Chalmers University of Technology, Gothenburg, Sweden. [34]BioInnovation Institute, Copenhagen, Denmark. [35]Department of Primary Care and Public Health, Imperial College London, London, United Kingdom. [36]AP-HP Pitié-Salpêtrière Hospital, Department of Pharmacology, Paris, France. [37]Department of Clinical Biochemistry, Rigshospitalet, University of Copenhagen, Copenhagen, Denmark. [38]Department of Cardiology, Rigshospitalet, University of Copenhagen, Copenhagen, Denmark. [39]Department of Medicine, University of Copenhagen, Copenhagen, Denmark. [40]Faculty of Health Sciences, University of Southern Denmark, Odense, Denmark. [41]Department of Clinical Physiology, Sahlgrenska University Hospital, Gothenburg, Sweden. [42]Biobyte Solutions GmbH, Heidelberg, Germany. [43]Molecular Medicine Partnership Unit, University of Heidelberg & EMBL, Heidelberg, Germany. [44]Biomedical Research Institute, Academy of Athens, Athens, Greece. [45]UK Dementia Research Institute, Imperial College London, London, United Kingdom. [46]MRC Centre for Environment and Health, Imperial College London, London, United Kingdom. [47]Université Paris Cité, Inserm U1124, Paris, France. [48]Victor Phillip Dahdaleh Institute of Genomic Medicine, McGill University, Montreal, Canada. [49]Department of Heath Research Methods, Evidence, and Impact, McMaster University, Hamilton, Canada. [50]McMaster Institute for Research on Aging, McMaster University, Hamilton, Canada. [51]Section of Genetics and Genomics, Imperial College London, London, United Kingdom. [52]Center for Genomic Medicine, Kyoto University Graduate School of Medicine, Kyoto, Japan. [53]Université Paris Cité, CNRS UMR 8251, Paris, France. [54]Center for Clinical Metabolic Research, Herlev–Gentofte Hospital, Copenhagen, Denmark. [55]Department of Clinical and Movement Neurosciences, University College London, London, United Kingdom. [62]These authors contributed equally: Kanta Chechi, Rima Chakaroun, Antonis Myridakis, Sofia K. Forslund-Startceva, Sebastien Fromentin. [63]Deceased: Peer Bork. *A list of authors and their affiliations appears at the end of the paper. ✉ e-mail: k.chechi@imperial.ac.uk; oluf@sund.ku.dk; stanislav.ehrlich@ucl.ac.uk; karine.clement@inserm.fr; m.dumas@imperial.ac.uk

## The MetaCardis Consortium

Rohia Alili [16], Chloe Amouyal [16], Karen Assmann [16], Ehm Astrid Andersson Galijatovic [12], Fabrizio Andreelli [16], Olivier Barthelemy [56], Jean-Philippe Bastard [57], Jean-Paul Batisse [56], Randa Bittar [58], Hervé Blottière [11], Frederic Bosquet [41], Rachid Boubrit [56], Olivier Bourron [41], Mickael Camus [11], Dominique Cassuto [16], Cécile Ciangura [16], Jean-Philippe Collet [56], Maria-Carlota Dao [16], Morad Djebbar [56], Angélique Doré [11], Line Engelbrechtsen [16], Soraya Fellahi [57], Leopold Fezeu [32],

**Philippe Giral**[59], **Agnes Hartemann**[41], **Bolette Hartmann**[12], **Gerard Helft**[56], **Jens Juul Holst**[12], **Marlene Hornbak**[12], **Andrea Rodriguez-Martinez**[1], **Jean-Sebastien Hulot**[36], **Richard Isnard**[56], **Sophie Jaqueminet**[11], **Niklas Rye Jørgensen**[41], **Hanna Julienne**[11], **Johanne Justesen**[12], **Judith Kammer**[5], **Nikolaj Karup**[12], **Mathieu Kerneis**[56], **Jean Khemis**[16], **Michael Kuhn**[31], **Véronique Lejard**[56], **Florence Levenez**[56], **Lea Lucas-Martini**[16], **Robin Massey**[56], **Nicolas Maziers**[11], **Jonathan Medina-Stamminger**[16], **Gilles Montalescot**[56], **Sandrine Moutel**[17], **Laetitia Pasero Le Pavin**[11], **Christine Poitou-Bernert**[16], **Francoise Pousset**[56], **Laurence Pouzoulet**[59], **Lucas Moitinho-Silva**[31], **Johanne Silvain**[56], **Mathilde Svendstrup**[12], **Timothy Swartz**[16,60], **Thierry Vanduyvenboden**[11], **Camille Vatier**[16], **Stefanie Walther**[5], **Eric Verger**[61] & **Aurélie Lampure**[61]

[56]Assistance Publique-Hôpitaux de Paris, Pitié-Salpêtrière Hospital, Cardiology Department, Paris, France. [57]Centre de Recherche Saint-Antoine, Sorbonne Université, INSERM UMR S938, Paris, France. [58]Assistance Publique-Hôpitaux de Paris, Pitié-Salpêtrière Hospital, Biochemistry Department of Metabolic Disorders, Paris, France. [59]Assistance Publique-Hôpitaux de Paris, Pitié-Salpêtrière Hospital, Endocrinology Department, Paris, France. [60]Integrative Phenomics, Paris, France. [61]Integromics Unit, Institute of Cardiometabolism and Nutrition, Paris, France.

