## [Transparent Peer Review file · Nature Communications]

A gut microbiome-kidney-heart axis predictive of future cardiovascular diseases.

Corresponding Author: Professor Marc-Emmanuel Dumas

Version 0:

Reviewer comments:

Reviewer #1

(Remarks to the Author)

The authors have adequately addressed this reviewer's concerns and suggestions. No further comments.

Reviewer #2

(Remarks to the Author)

The authors have addressed the concerns raised by the reviewers. This is excellent work highlighting the importance of diet and the gut microbiome on the development of CMD. I congratulate the team.

Reviewer #3

(Remarks to the Author)

The authors well addressed the comments from reviewer 2 and several other reviewers. One left issue to be further clarified is the validity of performing MR analysis from microbiome to phenotypes. A major issue in the field is there is no valid and replicable genetic instrument of gut microbiome, which means that MR analysis from gut microbiome to disease/phenotype is not reliable. It may be suitable to do the disease to gut microbiome MR analysis, but not the other way around. Therefore, the authors need to tone down this bi-directional claims throughout the manuscript and may not need to mention it in the abstract to avoid misleading readers.

Response to the reviewers

We thank the reviewers and the editor for reviewing our manuscript. Please see our answers to their comments (shown in blue) on a point-by-point basis below:

REVIEWERS' COMMENTS

Reviewer #1 (Remarks to the Author):

The authors have adequately addressed this reviewer's concerns and suggestions. No further comments.

Answer: Thanks so much.

Reviewer #2 (Remarks to the Author):

The authors have addressed the concerns raised by the reviewers. This is excellent work highlighting the importance of diet and the gut microbiome on the development of CMD. I congratulate the team.

Answer: Thanks so much.

Reviewer #3 (Remarks to the Author):

The authors well addressed the comments from reviewer 2 and several other reviewers. One left issue to be further clarified is the validity of performing MR analysis from microbiome to phenotypes. A major issue in the field is there is no valid and replicable genetic instrument of gut microbiome, which means that MR analysis from gut microbiome to disease/phenotype is not reliable. It may be suitable to do the disease to gut microbiome MR analysis, but not the other way around. Therefore, the authors need to tone down this bi-directional claims throughout the manuscript and may not need to mention it in the abstract to avoid misleading readers.

Answer: *We thank the reviewer once again and we completely agree that currently MR cannot be carried out reliably for gut microbiome related variables. This is exactly why we restricted our analyses to metabolites in the current manuscript and did not include any on the taxonomic or functional variables. Caveats still exist even with implementing MR on microbiome-related metabolites, which we have detailed in our methods section as well as pointed to in our limitations section. Additionally, we have now removed any direct reference to potential causality throughout the text including the abstract.*